# Assessing the impact of riverine water on the Northwest Pacific using normalized Total Alkalinity

Tatsuki Tokoro[1, 2], Shin-Ichiro Nakaoka[1], Shintaro Takao[1], Shu Saito[3, 4], Daisuke Sasano[4], Kazutaka Enyo[4], Masao Ishii[5], Naohiro Kosugi[5], Tsuneo Ono[6], Kazuaki Tadokoro[6], and Yukihiro Nojiri[1]

[1]currently at: Earth System Division, National Institute for Environmental Studies, Tsukuba, Japan.
[2]formaly at: Seto Inland Carbon-Neutral Research Center, Hiroshima University, Hiroshima, Japan.
[3]Administration Department, Japan Meteorological Agency, Tokyo, Japan.
[4]Atmosphere and Ocean Department, Japan Meteorological Agency, Tokyo, Japan.
[5]Department of Climate and Geochemistry Research, Meteorological Research Institute, Tsukuba, Japan.
[6]Fisheries Resources Institute, Japan Fisheries Research and Education Agency, Yokohama/Shiogama, Japan.

Corresponding author: Tatsuki Tokoro (tokoro.tatsuki@nies.go.jp)

**Abstract.** The impact of riverine water was assessed using salinity-normalized Total Alkalinity observations of the Northwest Pacific, including the coastal areas of Japan (20–50°N, 120–160°E). The observational data included surface carbonate parameters obtained from decades of surveys conducted by volunteer cargo ships and research vessels in this area. This study uses data and statistical methods (e.g., re-gridding and Fourier regression) like those in a previous study that analysed air-sea $CO_2$ flux but focuses instead on the diffusion of nTA from land. First, the seawater area affected by riverine water was identified using an Empirical Orthogonal Function analysis of normalized Total Alkalinity. The differences in normalized Total Alkalinity and Dissolved Inorganic Carbon from the surrounding area were then analysed to evaluate the potential drivers, such as riverine water supply, advection effects, and biological activities. In addition, the impact of riverine water on oceanic $CO_2$ uptake and acidification in the study area was assessed. The analysis showed that riverine water was the main cause of the higher total Alkalinity compared to the surrounding area, whereas its contribution to the increase in Dissolved Inorganic Carbon was relatively minor. The supply of riverine water had little effect on oceanic $CO_2$ uptake throughout the year. The supply of riverine water had a minor effect on pH but was a factor of coastal acidification in a calcification index. The supplied Total Alkalinity reduced this index change by 71% overall. The results of this study are expected to be further improved by enhancing observations, such as the vertical profiles of carbonate parameters, and are expected to expand to other sea areas and be applied to global budgets.

## 1. Introduction

The input of riverine water to the ocean is one of the most important flows in the Earth's system. In the carbon cycle, riverine water is a major carbon source for the oceans (Aufdenkampe et al., 2011; Bauer et al., 2013; Borges et al., 2005; Cai, 2011; Chen and Borges, 2009; Chen et al., 2013). Recent assessments indicate that the global carbon flux into the ocean via rivers is $1.02 \pm 0.11$ (Ave. $\pm$ SD) Pg-C yr$^{-1}$ (Liu et al., 2024). This is equivalent to about one-third of the oceanic absorption of atmospheric $CO_2$, which is the largest air-sea carbon flux to the surface ocean ($2.9 \pm 0.4$ Pg-C yr$^{-1}$, Friedllingstein et al., 2025). This riverine carbon flux is responsible for atmospheric $CO_2$ emissions and acidification in coastal areas (Carstensen and Durate, 2019; Duarte et al., 2013; Tranvik et al., 2009). However, strong carbon flows, such as biological pumps, impact carbonate distribution in coastal areas (Passow and Carlson, 2012; Regnier et al., 2013, 2022). For example, an increased biological pump in coastal areas will enhance $CO_2$ absorption from the atmosphere, but it will also promote anoxia in the sediments. In coastal areas, the mineralization of organic matter supplied by riverine water is expected to cause more severe acidification, which may stimulate harmful algal growth and adversely affect marine products such as bivalves (Fitzer et al., 2018; Kessouri et al., 2021; Wallace et al., 2014). Quantifying these complex flows is important for anticipating future climate change and ocean acidification as well as for projecting biogeochemical changes in coastal areas.

The Northwest Pacific Ocean, including the coastal areas of Japan, plays a crucial role in global carbon cycle due to its strong sink of atmospheric $CO_2$ (Takahashi et al., 2002, 2009). While several studies on carbonate systems have been conducted in the Northwest Pacific Ocean (e.g., Ishii et al., 2001; Murata et al., 1998; Takamura et al., 2010; Tokoro et al., 2023; Yoshikawa-Inoue et al., 1995, 2014), spatiotemporal variations in riverine water contribution in this region have not yet been quantified. Riverine water tracers (e.g., salinity, stable isotope, geogenic solutes like silica) are one of the most effective methods for evaluating the influence of riverine water. Although salinity is the most frequently assessed factor in riverine water transportation, other factors such as precipitation and evaporation also impact salinity. This can be a source of major error in the assessment of water transportation in the Northwest Pacific region, where high precipitation and evaporation by low-pressure systems and heat waves, respectively, have been observed (Kitamura et al., 2016; Miyama et al., 2021; Sugimoto et al., 2013).

The total Alkalinity (TA) normalized by salinity is a potential indicator to assess the influence of riverine water on ocean. TA is defined broadly as the charge difference between proton donors and acceptors (see Zeebe and Wolf-Gladrow, 2001 for a more detailed definition). TA is influenced by several factors, such as advection from different water masses and biological activity, including calcification and dissolution of calcium carbonate. Because TA is also highly correlated with salinity, TA normalized to a reference salinity (nTA) has been used to quantify the above-mentioned factors (e.g., Broecker and Peng, 1982; Lee et al., 2006; Millero et al., 1998). nTA is calculated as follows:

$$nTA = TA \cdot \frac{S_{ref}}{S} \tag{1}$$

where, $S$ and $S_{ref}$ are the measured and reference salinities (traditionally 35), respectively. Similarly, Dissolved Inorganic Carbon concentration (DIC), which is the sum of inorganic carbon species, was also normalized (nDIC). Equation (1) is

formulated based on the assumption that a water mass with zero salinity has zero TA, otherwise the right-hand side would go to infinity. However, this assumption is not true for riverine water because its TA is greater than zero, even when salinity is zero, owing to the weathering of carbonate and silicate rocks (e.g., Friis et al., 2003; Lehmann et al., 2023; Rassmann et al., 2016; Taguchi et al., 2009). Therefore, when Equation (1) is applied to areas affected by riverine water, the nTA value is higher than that of the surrounding seawater. Conversely, the influence of riverine water, along with factors such as water mass advection and biological activity, can be quantified by assessing the distribution of nTA. Unlike other methods that use salinity as a tracer for riverine water, nTA defined in Equation (1) excludes the effect of precipitation and evaporation, which is advantageous because contamination by water masses with almost zero salinity and TA can be excluded. Furthermore, nTA can be used to quantify the influence of seawater inflow from different local areas because distinct regional differences in nTA have been reported (Kakei et al., 2017; Lee et al., 2006; Takatani et al., 2014).

In this study, we aimed to analyse the influence of riverine water on the carbonate chemistry of the Northwest Pacific using nTA and other carbonate parameters measured by voluntary cargo ships and research vessels. First, spatiotemporal variations in the area in which riverine water significantly affected surface seawater carbonate chemistry were identified using an Empirical Orthogonal Function (EOF) analysis of the nTA distribution. Second, we focused on the differences in nTA and nDIC between the riverine water-affected area and surrounding unaffected areas. We quantified the riverine water supply and other contributing factors that affect nTA and nDIC. The final step involved the evaluation of the effects of riverine water on the environment in riverine water-affected area. Seawater $CO_2$ fugacity ($fCO_2$), pH and the aragonite saturation state of seawater ($\Omega_{arg}$) were the carbonate parameters that were used as the index of environmental changes caused by riverine water input. TA and DIC supplied by riverine water influence seawater $fCO_2$ and oceanic $CO_2$ uptake. Meanwhile, pH is an indicator of ocean acidification. However, $\Omega_{arg}$, which is the ratio of the concentration product of $[Ca^{2+}]$ and $[CO_3^{2-}]$ to the solubility product of aragonite, is another index of ocean acidification and has been reported to exhibit different spatiotemporal variations from pH (Kwiatkowski and Orr, 2018; Xue et al., 2021). Therefore, the analysis of changing pH and $\Omega_{arg}$ is expected to lead to a more detailed assessment of coastal acidification in the study area. This study also aimed to evaluate the effects of riverine water on future climate change and coastal acidification and to predict the effects of future environmental changes.

## 2. Methods

### 2.1 Data for analysis

The observational data in this study were collected by the National Institute for Environmental Studies (NIES), Meteorological Research Institute (MRI) of the Japan Meteorological Agency (JMA), and Japan Fisheries Research and Education Agency (FREA). The NIES data were produced as part of the Voluntary Observing Ship (VOS) programs for cargo ships (namely, M/S Alligator Hope, M/S Pyxis, M/S New Century 2, and M/S Trans Future 5). MRI, JMA, and FREA collected data from research vessels (R/V Mirai and R/V Hakuho-maru affiliated with Japan Agency for Marine-Earth Science and Technology

for MRI data; R/V Keifu-maru and R/V Ryofu-maru for JMA data; and R/V Wakataka-maru and R/V Soyo-maru for FREA data). These data were uploaded to the Surface Ocean $CO_2$ Atlas (SOCAT; Pfeil et al., 2013; Bakker et al., 2016, https://socat.info/index.php/data-access/) and Global Ocean Data Analysis Project (GLODAP, Key et al., 2015; Olsen et al., 2016; Olsen et al., 2020, https://glodap.info/). These observations were statistically processed by re-gridding, a second-order approximation of TA, carbonate equilibrium calculations, and Fourier regression into datasets with a spatial resolution of 1° × 1° and a temporal resolution of 0.1 year, as suggested in a previous study (Tokoro et al., 2023). Similar to the previous study, the data was excluded from the analysis if there was insufficient temporal data (n > = 60, corresponding to 6 years of data) in the 1°×1° spatial grid. Although the time interval (from January 1, 2000, to December 31, 2019), covered area (latitudes of 20–50°N and longitudes of 120–160°E), and original observations in the SOCAT by NIES were the same as those in the previous study, the original observations by MRI and JMA increased slightly (287750 to 291138) with the upgrade of the product (SOCAT ver. 2019 to 2023). The FREA dataset is a new addition to those used in previous studies. As the FREA data (n = 566175) were collected every 1 minute while the other data were collected every 10 min, the FREA data were weighted 1/10 with respect to the other data and weighted averaged as the gridded data. The GLODAP data for surface (<10 m depth) TA and DIC from the MRI and JMA were updated using ver. 2.2020 to 2.2023. The number of data points for JMA data increased from 2080 to 2163. However, the effect of increase in the data was small, which did not statistically affect the result (Table S1). Additionally, observations up to a depth of 150 m within the GLODAP were used to evaluate the effects of vertical advection (Text S1).

## 2.2 EOF analysis, Cause Analysis, and Environmental Impact Assessment

In this study, areas significantly affected by riverine water were identified by applying EOF analysis to the spatiotemporal variability of nTA. The EOF analysis breaks spatiotemporal variations into multiple orthogonal modes with multiple spatial patterns (principal EOF patterns) and time series (principal component time series) (e.g., Denbo and Allen, 1984). In addition to the effect of riverine water, the target area in this study is assumed to be subject to a mixture of multiple fluctuations, such as seasonal fluctuations of the Kuroshio and Oyashio flows. Compared to the direct use of carbonate parameters, the EOF analysis is expected to identify the influence of riverine water because the analysis is appropriate to quantify and separate multiple fluctuations. Using the principal EOF pattern and principal component time series with respect to the spatiotemporal variation of nTA, we identified the area significantly influenced by riverine water and labelled it as "Area A". We then labelled the surrounding region, with a rather similar surface area, as "Area B." The influence of riverine water was quantified as the value of Area A minus that of Area B (*dAB*) for all related parameters such as SST (Sea Surface Temperature), SSS (Sea Surface Salinity), nTA, and nDIC. Causal analysis of *dAB* of TA and DIC was performed using the following equation:

$$\frac{\partial dAB}{\partial t} = Sup_{rw} + C_{flux} + C_{res} \qquad (2)$$

The left-hand side of the equation represents the time derivative of $dAB$ of nTA and nDIC. $Sup_{rw}$ is the TA or DIC supply by riverine water; $C_{flux}$ is the term of difference in air-sea $CO_2$ flux between Areas A and B divided by the mixed layer depth (*MLD*). The MLD with the same spatiotemporal resolutions as the processed data (1° × 1° and 0.1 year) was calculated from the reanalysis of seawater temperature profiles by Japan Agency for Marine-Earth Science and Technology (JCOPE2M; Miyazawa et al., 2017, 2019, https://www.jamstec.go.jp/jcope/htdocs/distribution/). We determined an isothermal depth at ΔT = 0.2 °C with linear interpolation (de Boyer Montégut et al., 2004; Holte and Talley, 2009). $C_{flux}$ was applied only to the DIC input. $C_{res}$ is the TA or DIC input due to other residual factors related to $dAB$ (e.g., horizontal and vertical advection and biological activity).

$Sup_{rw}$ was estimated from river discharge and TA or DIC concentrations in river water as follows:

$$Sup_{rw} = \frac{Flow_r \cdot C_r}{MLD \cdot A} \tag{3}$$

where, $Flow_r$ is the monthly total flow rate of the river, $Cr$ represents the TA or DIC concentrations in the associated rivers, $A$ is the area of Area A. The flow rate was estimated using the Water Information System database of the Ministry of Land, Infrastructure, Transport, and Tourism of Japan (http://www1.river.go.jp/). As TA and DIC data were not available for all rivers, the referential values for zero-salinity endmembers in the three most river-influenced inner bays (Tokyo Bay, Ise Bay, and Osaka Bay) in the relevant coastal areas were used (518–1006 and 475–1371 µmol kg$^{-1}$ for TA and DIC, respectively; Taguchi et al., 2009; Tokoro et al., 2021). The DIC range was determined based on the maximum (1171 µmol kg$^{-1}$ in Tokyo Bay) and minimum (675 µmol kg$^{-1}$ in Ise Bay) freshwater endmember values among the three bays, with an additional ±200 µmol kg$^{-1}$ to account for seasonal variation estimated in the previous study (Tokoro et al., 2021). Although the TA range does not account for the seasonal variation, the range of TA variation in global rivers due to water temperature fluctuations is reported to be less than 10 µmol kg$^{-1}$ (Romero-Mujalli et al., 2019). This range is one order of magnitude smaller than that among the three bays described above, thus it was decided that the seasonal variation could be disregarded in the analysis. All data were regridded to the resolution of the processed SOCAT and GLODAP data (spatial and temporal resolution of 1° × 1° and 0.1 year, respectively).

Air-sea $CO_2$ flux ($F$) was determined as follows:

$$F = k \cdot K_0 \cdot (fCO_{2water} - fCO_{2air}) \tag{4}$$

where, $k$ is transfer velocity defined by the wind speed at 10 m above the sea surface (Wanninkhof, 2014). $K_0$ is the solubility of $CO_2$ in seawater and was calculated using an empirical equation based on SST and SSS (Weiss, 1974). $fCO_{2water}$ and $fCO_{2air}$ are the fCO$_2$ in surface seawater and air, respectively. $C_{flux}$ was determined by the difference in air-sea $CO_2$ flux between Areas A and B, which was calculated from the processed fCO$_2$ data and wind velocity from the database of the Cross-Calibrated Multi-Platform (CCMP, Atlas et al., 2011; Mears et al., 2019, https://data.remss.com/ccmp/v02.0/, version 2.0). The details of the calculations are the same as those used in a previous study (Tokoro et al., 2023).

The effect of riverine water on air-sea $CO_2$ flux and acidification was quantified by multivariate analysis using $dAB$ of seawater fCO$_2$, pH and $\Omega_{arg}$ as the objective variables and $dAB$ of SST, SSS, nTA, and nDIC as the explanatory variables. The pH was

determined as total scale from $fCO_2$ and TA, and $\Omega_{arg}$ was calculated using the equation of an aragonite solubility product (Mucci, 1983) and the concentration of $Ca^{2+}$ and $CO_3^{2-}$. Both pH and $\Omega_{arg}$ were estimated using the CO2SYS program (Lewis and Wallace, 1998) and the "recommend" coefficients in a literature (Zeebe and Wolf-Gladrow, 2001). These Seawater $fCO_2$, pH, and $\Omega_{arg}$ can be unambiguously determined from equilibrium calculation using SST, SSS, TA, and DIC. However, it is difficult to intuitively understand the contribution of the explanatory variables because these have a non-linear relationship. Therefore, we considered that a multivariate linear model should be useful in this study. Partial least squares regression (PLS regression; Wold et al., 2001) was used in the multivariate analysis to prevent multicollinearity, particularly considering the strong correlation between SSS and nTA and nDIC, due to riverine water having high nTA and nDIC.

### 3. Results

Figure 1 represents the processed spatial distributions of SSS, TA, and nTA. The SSS showed a north-south gradient, which was attributed to the high-salinity Kuroshio current and the relatively low salinity Oyashio current. There was also an area of reduced SSS along the Pacific coast of mainland Japan (32–34°N, 132–140°E; Figure 1a). These trends were similar for TA, which exhibited high correlation with SSS ($R^2 = 0.94$) (Figure 1b, d). The nTA was high in the northern part of the study area, and slightly high values were also observed along the Pacific coast of Japan (Figure 1c). The intercept of the regression line between SSS and TA was $528.04 \pm 461.09$ µmol kg$^{-1}$ (Ave ± SD), which was consistent with the TA value of large river on the continental side like the Amur River (589 µmol kg$^{-1}$; Andreev and Pavlova, 2009) and the range of riverine water used in this study (518–1006 µmol kg$^{-1}$) (Figure 1d). Because the intercept value was above zero, nTA seemed to be inversely proportional to SSS ($R^2 = 0.57$), with a lower SSS tending to have a higher nTA (Figure 1e). These results indicate that the study area was affected by freshwater with TA above zero, especially in the northern area and on the Pacific coast of Japan, and that nTA could be used as a tracer for freshwater inputs.

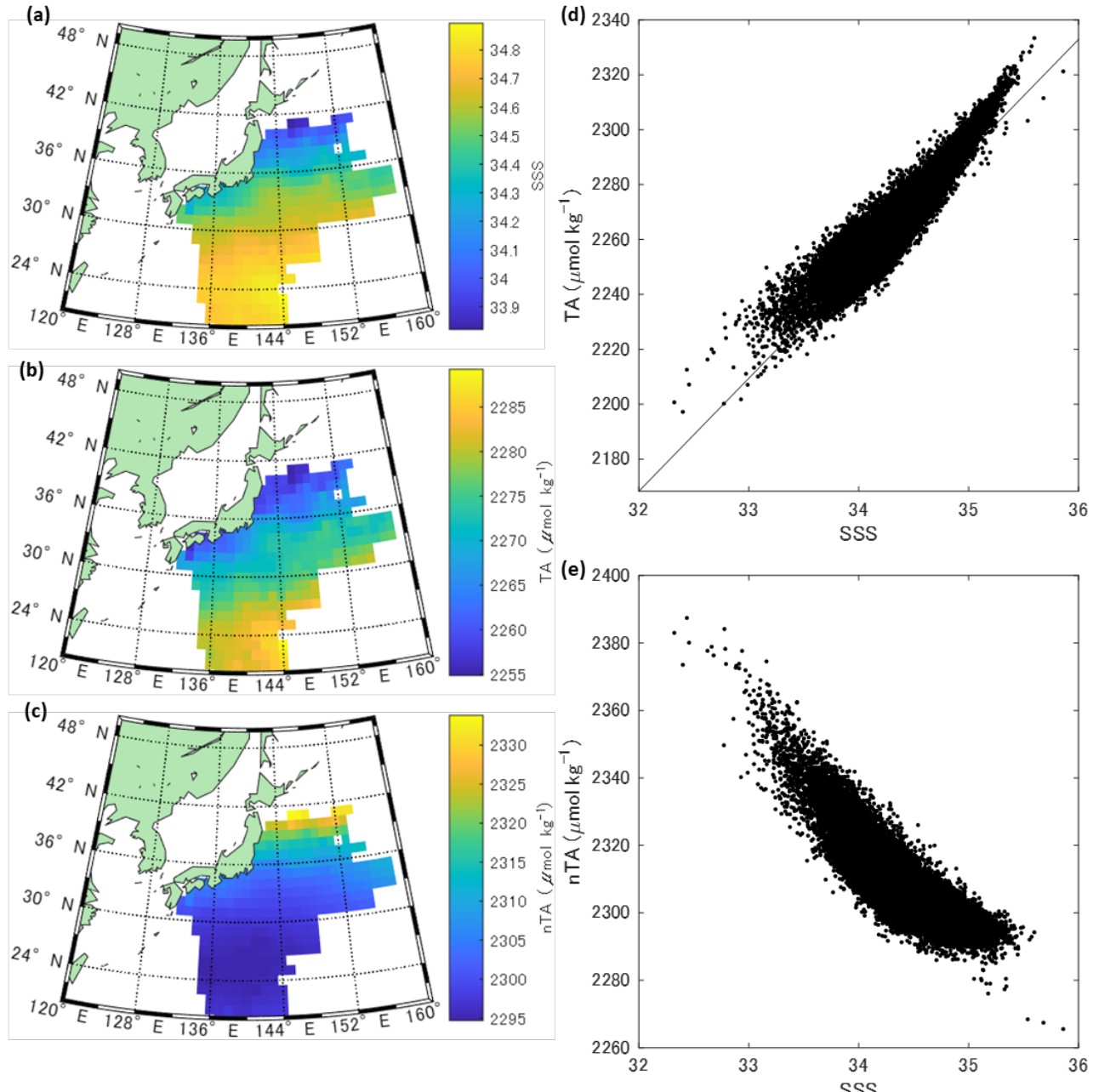

**Figure 1 (a–c): Spatial distribution of (a) mean Sea surface Salinity (SSS) (b) Total Alkalinity (TA) and (c) normalized TA (nTA).**
**The rectangular in (a) show the reduced SSS area along the Pacific coast of mainland Japan (32–34°N, 132–140°E) (d and e):**
**Scatterplot of (d) SSS-Total Alkalinity (TA) and (e) SSS-nTA. The black line is the linear regression line calculated using the least**
**squares method between SSS and TA ($R^2$ = 0.94, p<0.05, TA = (50.46 ± 0.06) × SSS + (528.04 ± 461.09) (Ave. ± SD)).**

The spatial distribution of nTA in the three most dominant modes of the EOF analysis is shown in Figure 2(a–c). The principal EOF patterns were indicated by a parameter defined as "EOF anomaly," denoted by red or blue in Figure 2a. The spatiotemporal variation of the nTA anomaly in each mode was expressed as the product of the EOF anomaly and the principal component time series. In the most dominant mode (Mode 1), the principal patterns exhibited a clear north-south difference at approximately 37°N (Figure 2a). The principal component of the time series in Mode 1 (Figure 2d) indicated that the annual cycle was predominant. Annual maximum of nTA was observed in summer in the area south of 37°N, while it was in winter in the area north of 37°N (data not shown). The seasonal variation south of 37°N can be explained by the fact that riverine water supply is proportional to the precipitation over land, thereby reaching its peak in summer on the Pacific side of Japan (Database of Japan Meteorological Agency, https://www.data.jma.go.jp/stats/etrn/index.php). For the area north of 37°N, this may be due to the southward transportation of high-nTA surface seawater in the Pacific subarctic region by north winds in winter or winter vertical mixing that supplied subsurface high-nTA seawater to the surface. In addition, the EOF anomaly south of 37°N was significantly correlated with distance from the Japanese mainland (Figure 2e). Because EOF anomaly is a parameter indicating the strength of annual cycle fluctuation in nTA, as shown in Figure 2d, Figure 2e indicates that the degree of temporal variation in nTA increased significantly closer to mainland Japan in this mode. The other major modes (Modes 2 and 3; Figure 2b, c) had an east-west distribution north of 37°N. A distribution such as this could represent seasonal or multi-year variations in the flow path of the Kuroshio Extension. After mode 4, the variation north of 37°N was also explained, and thus mode 1 was dominant for the variation south of 37°N.

Based on the above considerations, we determined that it was appropriate to use the Mode 1 EOF anomaly south of 37°N as an indicator of the influence of riverine water from mainland Japan. Although the area north of 37°N contains Japanese rivers with large flow rates (e.g., the Kitakami River), their influence was excluded from the analysis in this study because the influence of high nTA fluctuation in the subarctic gyre was too dominant to extract the influence of riverine water from Japan. The influence of subarctic Pacific seawater can be expected even around 37°N; however, this is not expected to have a significant effect on the statistics in Area A, such as the spatial mean values.

Area A was defined to be south of 37°N and within 250 km from land (Figure 2f). This distance was determined using change-point detection (Killick et al., 2012), indicating that the mean EOF anomaly south of 37°N changes abruptly before and after this distance. The outer edge of Area A roughly aligned with the Kuroshio axis, as indicated by the JCOPE2 SSH data (SSH = 0.2 m). Although the EOF anomaly outside Area A also shows a relatively weak correlation with distance, this correlation is thought to be due to the influence of riverine water from areas other than Japan, such as East Asia, which was difficult to quantify using the measurement data in this study.

To minimize the influence of riverine water outside Japan and differences in latitude and longitude, Area B was determined as adjacent to Area A, similar in size to the area for comparison (Figure 2f).

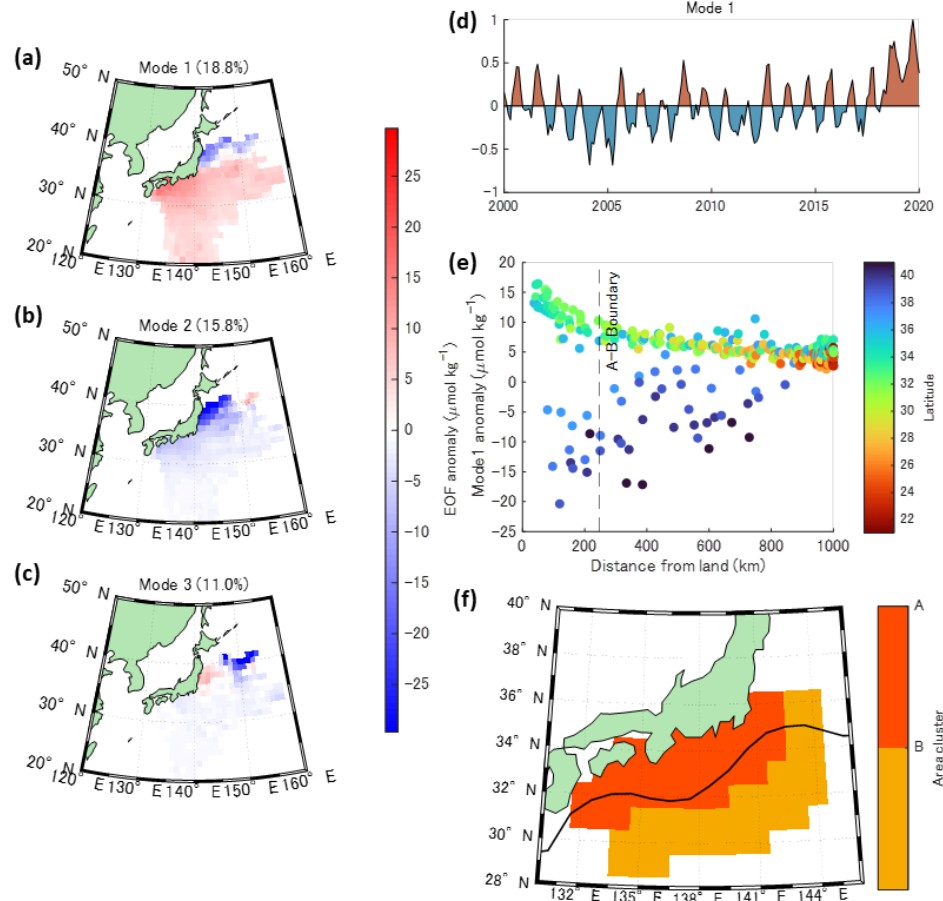

**Figure 2. (a–c): EOF anomaly of (a) Mode 1, (b) Mode 2, and (c) Mode 3. The percentage in the bracket (18.8%) indicates the contribution of this mode to the original variation. (d): Principal component of time series of Mode 1. (e): Scatterplot of EOF anomaly of Mode 1 versus distance from Japan mainland. The plot colours indicate the latitude. The dotted line represents the boundary of Areas A and B estimated using the changepoint analysis. (f): Distribution of Area A and B. The black curve indicates mean Kuroshio axis during the whole period in this study (2000-2019).**

The monthly and annual average values of the differences in the relevant parameters (SST, SSS, nTA, and nDIC) between Areas A and B ($dAB$) are shown in Figure 3. The average values for each parameter in Areas A and B are summarized in Table S2. All $dAB$ parameters showed seasonal variation, and the absolute values of $dAB$ were largest in winter for SST and nDIC and in summer for SSS and nTA. On a decadal scale, the absolute values of all $dAB$ tended to increase significantly. This indicates an increase in the supply of riverine water, which is consistent with the Japan Meteorological Agency's report (https://www.data.jma.go.jp/cpd/cgi-bin/view/index.php) that precipitation in relevant areas of Japan increased during the period analysed in this study.

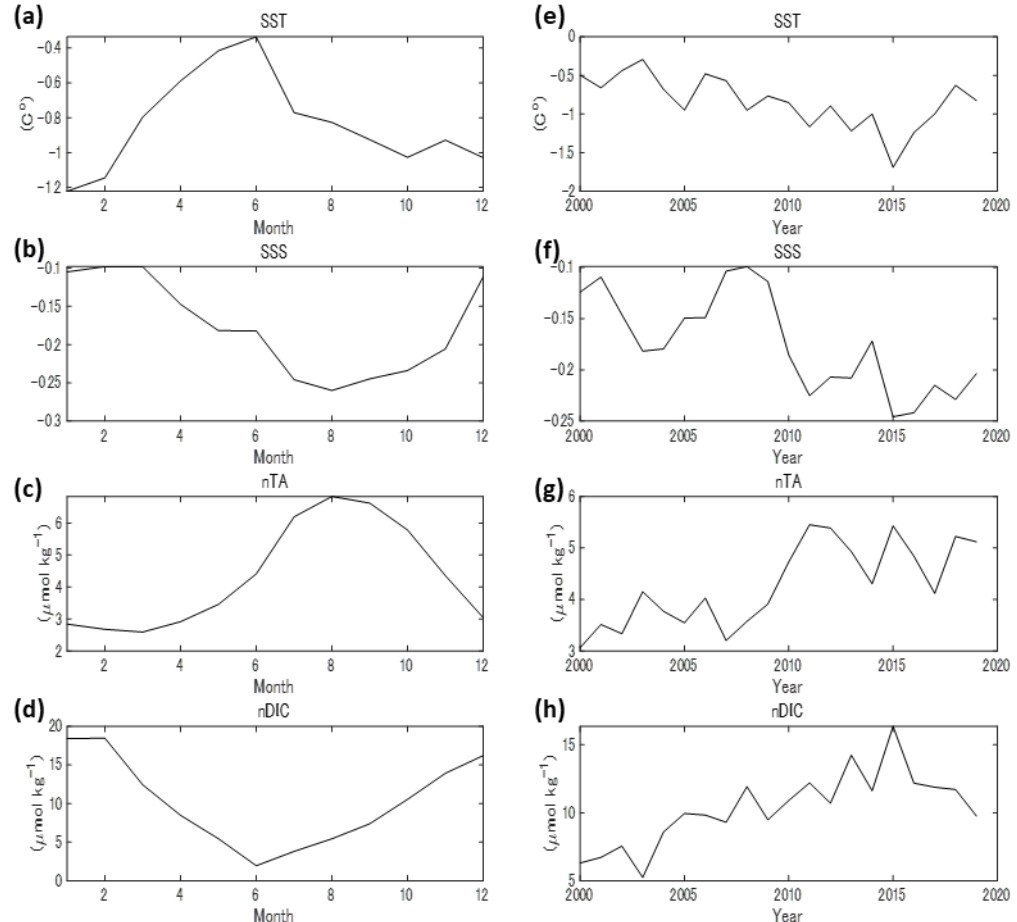

**Figure 3. The differences in Sea Surface Temperature (SST), Sea Surface Salinity (SSS), normalized Total Alkalinity (nTA), and normalized Dissolved Inorganic Carbon (nDIC) between Areas A and B (*dAB*). (a-d): Seasonal variations. (e-h): Annual variations. Although the time step in this study was 0.1 year, the monthly values were calculated by spline interpolation of values at 1/12-year intervals. The same applies to the other figures.**

Figure 4 represents the time-series variations of each term in Equation (2) used for the causal analysis of the *dAB* of nTA and nDIC. The river discharge for $Sup_{rw}$ was calculated as the sum of the 37 rivers bordering Area A (see Text S2). The maximum $Sup_{rw}$ for both nTA and nDIC were observed during the summer months, when precipitation on the Pacific coast of Japan was the highest (Figure 4b and 4d). Though the air-sea $CO_2$ flux term ($C_{flux}$) also showed a clear seasonal variation, the annual

average did not. The residual term ($C_{res}$) tended to be negative for both nTA and nDIC; however, the seasonal patterns and ranges of variation differed considerably.

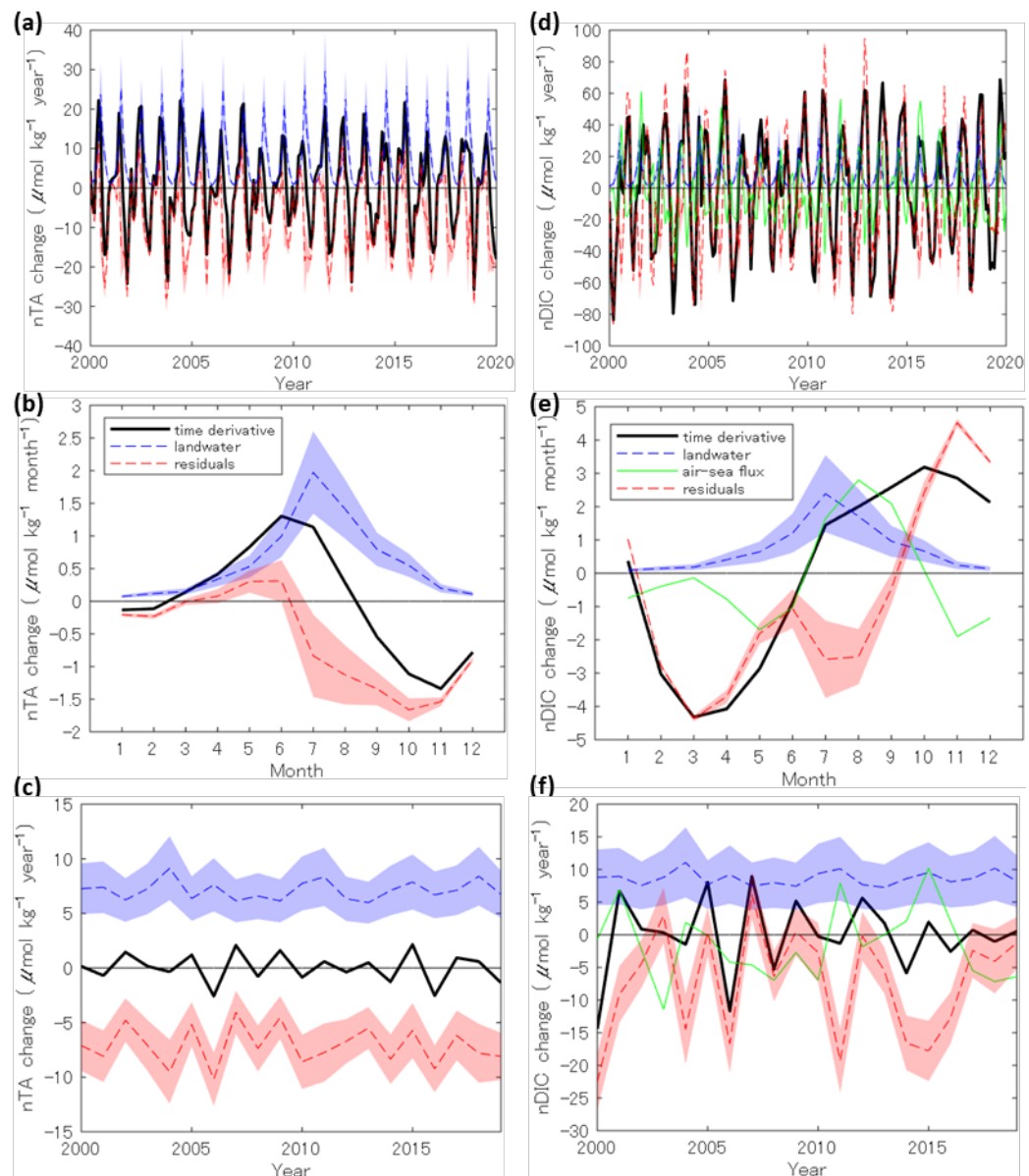

**Figure 4. (a): Temporal variations of normalized Total Alkalinity (nTA, a-c) and normalized Dissolved Inorganic Carbon (nDIC,**
**d-f) time derivatives and related terms in Equation (2). Panels show (a,d) temporal variations, (b, e) monthly averages, and (c, f)**
**yearly averages. Shaded areas indicate error ranges calculated from the upper and lower limits of riverine Total Alkalinity (TA,**
**518-1006 µmol kg⁻¹) and Dissolved Inorganic Carbon (DIC, 475-1371 µmol kg⁻¹); these are not random errors, so the same ranges**
**apply to monthly and yearly averages. Dotted lines within shaded areas represent the average values.**

The PLS regression showed that the explanatory variables SST, SSS, nTA, and nDIC explained well the objective variables seawater $fCO_2$ ($r^2 = 0.996$), pH ($r^2 = 0.993$), and $\Omega_{arg}$ ($r^2 = 0.996$) (Figure 5). The average contributions due to each explanatory variable in the PLS analysis were consistent with those in the equilibrium calculation for seawater $fCO_2$, pH and $\Omega_{arg}$ using the averages in Area B and the respective $dAB$ values (Table S3). The seawater $fCO_2$ and $\Omega_{arg}$ in Area A were lower than those in Area B by -3.61 $\pm$ 0.70 $\mu$ atm and -0.09 $\pm$ 0.01 (Ave. $\pm$ SE), respectively. Furthermore, the pH was comparable to that in

Area B ($+3.7 \pm 0.7$) $\times$ $10^{-3}$. All parameters showed clear seasonal variations.

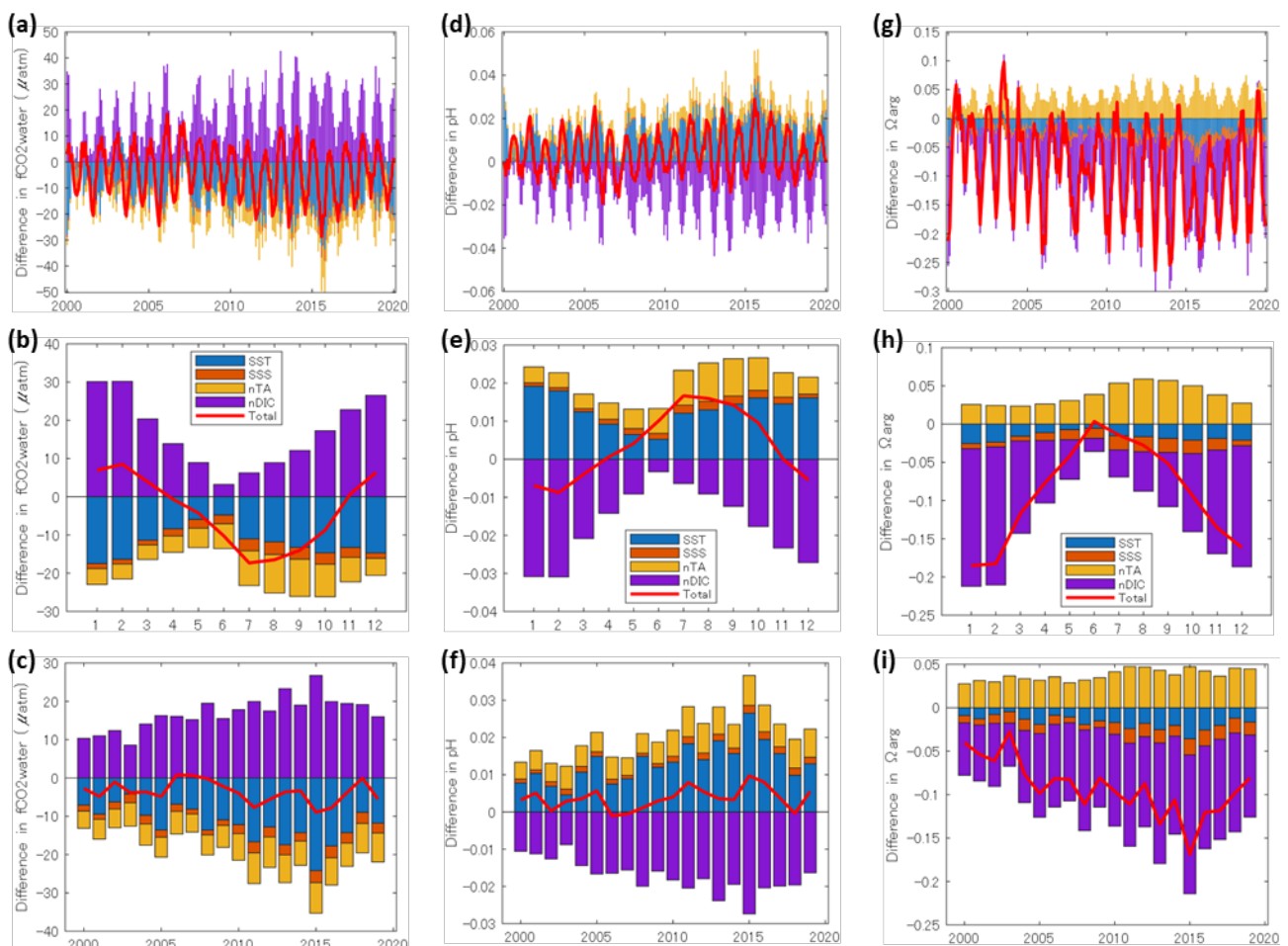

Figures 5. (a): Temporal contributions of Sea Surface Temperature (SST), Sea Surface Salinity (SSS), normalized Total Alkalinity (nTA), and normalized Dissolved Inorganic Carbon (nDIC) in the PLS analysis to the difference between Area A and B in (a-c)
seawater $CO_2$ fugacity ($fCO_2$), (d-f) pH, and (g-i) aragonite saturation state ($\Omega_{arg}$). Panels show (a, d, g) the temporal contributions, (b, e, h) monthly averages, and (c, f, i) yearly averages. Red lines indicate the sum of contributions, which closely matches the temporal variation of the differences ($dAB$) in each variables.

## 4. Discussion

The extent of Area A defined using nTA (Figure 2f) was consistent with the low-salinity area of the Pacific coast of Japan (Figure 1a). The SSS south of 37°N exhibited an abrupt change in value 210 km from mainland Japan (Figure S4), which is consistent with the distance of the boundary between Areas A and B (250 km). Because SSS is affected by precipitation and evaporation, the distance defined by nTA is considered more accurate as the boundary for Area A. However, the consistency between the two distances provides is one of the strong evidence that the nTA is an effective indicator for riverine water. Because these distances were calculated statistically using 20 years of observational data on SSS and nTA. it would be challenging to obtain a similarly significant result from single observations. Therefore, the findings of this study, which quantify the extent of the influence of riverine water, are novel and offer new insights.

To evaluate the effect of the Kuroshio Current on riverine water distribution, another EOF analysis was performed using data from the Kuroshio Large Meander period (2017–2020 in this study) in which the Kuroshio path followed an alternative meandering path south of its usual course (Kawabe, 1985). Area A was within the range of 350 km from land, utilizing the Mode 1 EOF anomaly during the meander period, whereas it was 250 km during the entire period. This result was consistent with the observed Kuroshio path meandering southwards, that is, away from the Pacific coast of Japan, and supports the assumption that the Kuroshio Current effects Area A. However, Area A extended further out of the open ocean than the Kuroshio axis during the entire study period (Figure 2f). Therefore, the Kuroshio Current limited the spread of riverine water to some extent but did not completely inhibit it.

The supply of TA by riverine water was higher than the time deviation of $dAB$ for almost the entire period, suggesting that the supply was one of the main factors contributing to the increase in $dAB$ (Figure 4). On the other hand, the annual rate of increase in the $dAB$ of nTA was almost zero (-0.02 ± 0.05 µmol kg$^{-1}$ year$^{-1}$). Therefore, the residual term would be the negative value of the same scale with $Sup_{rw}$ term for TA. Horizontal advection is assumed to be the primary cause of the negative residual term. This effect can be estimated using the product of the current velocity and the $dAB$ of nTA. The fact that the period of increasing $dAB$ of nTA (Figure 3c) coincides with the period of increasing residual term absolute value (Figure 4b) supports this assumption. On the other hand, the effect of vertical advection estimated from the vertical gradient of nTA and mixed layer depth was not significant (see Text S1). Other causes of the residual term are differences in biological activity, such as calcification, nitrate consumption, and organic Alkalinity. Calcification by coccolithophores is one of the most important process driving changes in oceanic nTA. However, no regional difference in calcification rate of up to 10 µmol kg$^{-1}$ year$^{-1}$ (Figure 4b) has been reported between the Japanese coastal area (Area A) and the surrounding sea area (Area B) (Hopkins and Balch, 2018; Krumhardt et al., 2019). Another component of TA include nitrate, which is consumed during photosynthesis to raise nTA (Brewer and Goldman, 1976). The data on total nitrate (TN) concentration is available in GLODAP, and the mean values were 1.19 ± 0.11 µmol kg$^{-1}$ and 0.34 ± 0.04 µmol kg$^{-1}$ in Area A and B, respectively. As the monthly difference were

largest in the winter (2.11 µmol kg$^{-1}$) and almost reach zero in the summer, the main source of nitrate would be vertical advection rather than input from land. Nonetheless, the effect of nitrate on TA was probably one order of magnitude smaller than that of the riverine water supply and is negligible as the main source of nTA. The inflow of organic Alkalinity is assumed to be another component influencing nTA fluctuations (Cai et al., 1998; Song et al., 2020; Yang et al., 2015). Organic Alkalinity is the TA component that originates from organic acids such as humic acid, found in terrestrial sources like wetlands

and coastal sediments. Although no data on organic Alkalinity from mainland Japan could be found, a reference on organic Alkalinity from rivers in southern coast of China reports that the maximum value for organic Alkalinity was 18 $\mu$ mol kg$^{-1}$ under condition of salinity exceeding 30 like the analysis in this study (Song et al., 2023). Assuming the outflow of organic Alkalinity from mainland Japan is of a similar order of magnitude, this represents less than 1% of the oceanic TA. Furthermore, compared to the upper and lower limits of the TA concentrations in the associated rivers (518-1006 µmol kg$^{-1}$), this is a

sufficiently small. Therefore, in this analysis, the influence of organic Alkalinity was considered negligible. However, it will have a significant impact on future TA budget analysis conducted under lower salinity conditions or in environments with abundant wetlands, such as those found in northern Japan, including Hokkaido.

        The DIC variation is more complicated than the TA variation because the spatiotemporal variation in biological activity and vertical advection is larger and more significant than that of TA, in addition to the effect of $CO_2$ exchange with the atmosphere.

Unlike the TA, the seasonal variations in the riverine water supply were not consistent with those in the time derivative (Figure 4e). This trend indicated that riverine water supply was not the primary driver of DIC variation in Area A. The effect of oceanic $CO_2$ uptake was highly variable and unclear on a decadal scale. However, on a seasonal scale, it was more distinct, resulting in negative shifts in the summer residual terms and positive shifts in the winter residual terms. This seasonal variation indicated that the oceanic $CO_2$ uptake in Area A was larger than that in Area B in summer, and vice versa in winter. Consequently, the

residual term of nDIC indicated a strong and complex seasonal variation, which included a maximum in the winter and two minimums in spring and summer with a difference of 100 µmol kg$^{-1}$ year$^{-1}$. Contrary to TA, DIC was affected by nutrient and organic matter loading from terrestrial sources, and was strongly affected by differences in biological activity between Areas A and B. Furthermore, a significant trend in the vertical profile of DIC was observed (Text S1). Consequently, some of the residual terms may have been affected by vertical advection.

To quantify the effect of biological activity, the effects of horizontal and vertical advection were estimated based on the following assumptions: 1) the residual term for nTA ($C_{res\_TA}$) was assumed to be equal to the horizontal advection term, based on the considerations that the effects of vertical advection and other effects on nTA are negligible. 2) The horizontal advection terms for nTA and nDIC were proportional to $dAB$ of nTA ($dAB_{TA}$) and nDIC ($dAB_{DIC}$), respectively. Several previous studies have used normalized values for the calculation of TA and DIC advection (Broecker and Peng, 1992; Keeling and Peng, 1995)

and have also proven their validity as an approximation (Robbins, 2001). Because the advection term is calculated as the product of concentration gradient and flow field, and the associated distance and flow velocity are the same at $dAB_{TA}$ and $dAB_{DIC}$, it can be assumed that the horizontal advection effect on DIC ($C_{hadv\_DIC}$) can be estimated using the following equation:

$$C_{hadv\_DIC} = K \frac{dAB_{DIC}}{D_{AB}}$$

$$C_{res\_TA} = K \frac{dAB_{TA}}{D_{AB}}$$

$\quad C_{hadv\_DIC} = C_{res\_TA} \cdot \frac{dAB_{DIC}}{dAB_{TA}} \quad$ (5)

where, $K$ and $D_{AB}$ are the index values associated with the current field and horizontal distance between Areas A and B, respectively. These parameters were assumed to be the same for TA and DIC calculations. 3) Vertical advection for nDIC ($C_{vhadv\_DIC}$) was estimated from mixing with subsurface water when the mixed layer was deepened. Specifically, we used a simplified equation of the method of a previous study (Ishii et al., 2001).

$\quad \int C_{vadv_{DIC}}(t)dt = \frac{\{\Delta nDIC(t+1)+\Delta nDIC(t)\}}{2} \times \frac{\{MLD(t+1)-MLD(t)\}}{\{MLD(t+1)\cdot\rho_{MLD}(t+1)\}}$

$$\Delta nDIC(t) = \{nDIC(July, MLD(t)) - nDIC(t, MLD(t))\} \cdot \rho_{MLD}(t) \qquad (6)$$

where, $\Delta nDIC(t)$ is the difference between the average nDIC in July ($nDIC(July)$) and the nDIC at time $t$ at mixed layer depth ($MLD(t)$) multiplied by the seawater density at $MLD(t)$ ($\rho_{MLD(t)}$). The nDIC profile at a depth below the mixed layer was assumed to be maintained at its average value in July; as the mixed layer deepened, the difference from the July profile was

added to the nDIC in the mixed layer. The effect of vertical advection on nDIC was negligible when the mixed layer depth did not change or became shallower. The details on the calculation of the nDIC profiles are provided in Text S1.

The effects of horizontal and vertical advection on nDIC are shown in Figure 6. However, due to the large error range in the vertical advection term for each time step (the median was 137%), only the monthly and annual averages are shown in the figure. The horizontal and vertical advection effects were identified mainly from autumn to winter with an annual mean value

of -15.55 ± 1.35 μmol kg$^{-1}$ year$^{-1}$ and 15.55 ± 2.25 μmol kg$^{-1}$ year$^{-1}$, respectively (Ave ± SE). Seasonally, the residual term had a maximum in the winter and two minimums in spring and summer, with an overall mean of -7.30 ± 3.24 μmol kg$^{-1}$ year$^{-1}$.

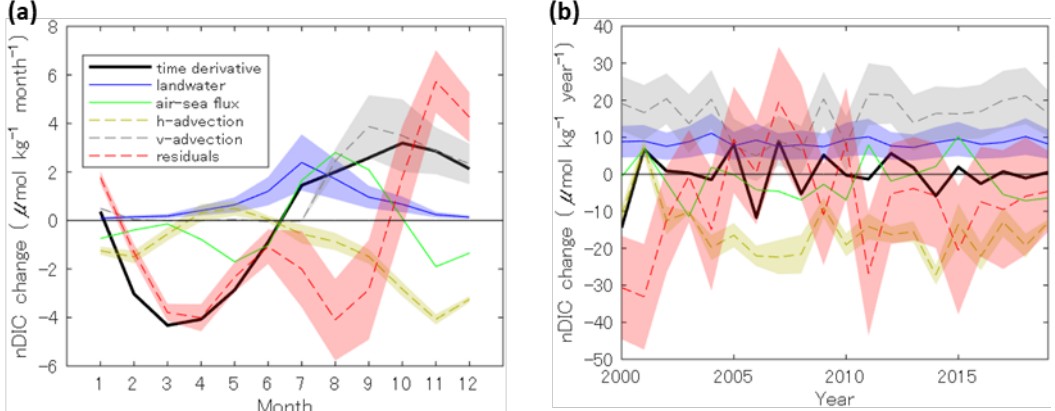

**Figures 6. Temporal variations of normalized Dissolved Inorganic Carbon (nDIC) time derivative and related terms in Equation (2)**

**(a): The monthly averages. (b): The yearly averages. The shaded areas of horizontal advection (h-advection) and vertical advection**

**(v-advection) are the random error calculated using the error propagation (see Text S1). Therefore, unlike the error ranges of riverine supply ($Sup_{rw}$), these were reduced by $10^{-0.5}$ or $20^{-0.5}$ by monthly or yearly averaging, respectively.**

Although the above results were not definitive because of many assumptions and large error ranges, the fact that the final residual term on the decadal scale was almost negative supports the idea that Area A is likely under heterotrophic conditions within the mixed layer. The positive shift in the residual term before and after 2007 was due to the low vertical advection term caused by the small MLD during winter. The 2007 minimum SST in Area A (19.41 °C) was the highest of the overall annual minimum SST (17.71 ± 0.66 °C, Ave ± SD), and thus the surface seawater mixing would be weaker than that in other years. Therefore, the residual term for this period does not necessarily indicate changes in biological activity. However, the high SST during winter suggests that the decomposition of biofixed carbon may be accelerated. Seasonally, the March-April minimum in the residual term was assumed to be the result of phytoplankton blooms in the open ocean area in Area A. Another minimum in July-August was assumed to be due to the primary production of phytoplankton under eutrophic conditions in the inner bays and near-shore areas. The winter maximum can be explained by the upwelling of organic matter and the decomposition of bio-fixed carbons within the mixed layer.

It should be noted that the DIC residual term includes the effects of–the water $CO_2$ flux in near-shore areas. However, unlike the air-sea $CO_2$ flux in the ocean ($C_{flux}$ in Equation (2)), the air-water $CO_2$ flux in near-shore areas is very difficult to quantify because observations of coastal $fCO_2$ and the physical regulating factor defined as "transfer velocity" (e.g., Wanninkhof, 2014) are limited. The estimation of this $CO_2$ flux varies widely among the existing studies. For example, global average models (Aufdenkamp et al., 2011; Tranvik et al., 2009) have estimated that approximately half of the carbon supply from land is released into the atmosphere via the air-water $CO_2$ flux in near-shore areas. However, the inner bays affected by riverine water in this study showed a trend in atmospheric $CO_2$ absorption (Tokoro et al., 2021), that is unlikely to follow the global average trend. Therefore, enhanced observation and analysis of atmospheric $CO_2$ exchange in nearshore areas is essential for a more accurate assessment of the residual term as an indicator of biological activity.

Despite Area A having a higher nDIC than Area B (Figure 3), seawater $fCO_2$ in Area A tended to be lower than in Area B (-3.61 ± 0.70 μatm). This was mainly because of the lower SST and higher nTA in Area A than those in Area B (Figure 5). The low SST can likely be attributed to riverine water, which is mainly supplied by snowmelt and rainfall from the mountains and highlands and therefore tends to have lower original water temperatures. In particular, the short flow paths of Japanese rivers can likely limit the effects of heating due to solar radiation and other factors. The three inner bays with strong river influence had lower average water temperature (18.66–19.23 °C, Tokoro et al., 2021) than areas A (22.02 °C) and B (22.86 °C), which support our assumption. The decrease in $dAB$ of seawater $fCO_2$ peaked in the summer (-15.90 ± 4.72 μatm from July to September), when the decrease in nDIC due to biological activity and the increase in nTA due to river supply coincided. However, this decreasing in seawater $fCO_2$ had little effect on annual oceanic $CO_2$ uptake in Area A. Compared to the hypothetical case where riverine water did not affect seawater $fCO_2$ (seawater $fCO_2$ in Area A were the same as that in Area B), the change in air-sea $CO_2$ flux would be similar (0.00 ± 0.33 mol m$^{-2}$ year$^{-1}$). This was because the decrease in air-sea $CO_2$

flux in summer almost offsets the increase in winter. Although the increase in seawater $fCO_2$ in winter (+7.25 ± 1.12 µatm from December to February) was smaller than the decrease in summer (-15.90 ± 4.72 µatm from July to September), the air-sea $CO_2$ fluxes in summer and winter were coincidentally balanced, owing to higher wind speeds in winter.

In the evaluation of acidification, different trends were observed for pH and $\Omega_{arg}$. While pH showed little difference between Areas A and B ((+3.7 ± 1.4) × $10^{-3}$), $\Omega_{arg}$ revealed a clear acidification trend in Area A. Regarding pH, the acidification trend
in Area A caused by nDIC increase appears to be offset by the above-mentioned low SST effect and the nTA inflow (Figure 5). Meanwhile, for $\Omega_{arg}$ seawater in Area A was notably more acidified than in the surrounding sea area, based on a 20-year average (-0.09 ± 0.01). This result is an example where pH and $\Omega_{arg}$ exhibit different behaviours, demonstrating that acidification by riverine water has little effect on hydrogen ion concentration, but exerts a significant influence on biological activities such as calcification. A similar trend is expected to occur in coastal areas north of the temperate zone where riverine
water cooling takes place. Although the inflow of nTA has a small effect on pH (+0.01), nTA was only mitigation factor for $\Omega_{arg}$ among the explanatory variables, and reduced acidification due to other factors to 71% (-0.13 to -0.09). Although no significant decadal change in pH of *dAB* was observed regarding acidification, coastal acidification indicated by $\Omega_{arg}$ was found to be progressing, with *dAB* of $\Omega_{arg}$ decreasing by -0.04 ± 0.01 per decade. This can be attributed to an increasing trend in *dAB* of nDIC (0.32 ± 0.08 µmol kg$^{-1}$ year$^{-1}$) due to the trend of increasing precipitation in Japan.


## 5. Conclusions

In this study, we identified areas affected by riverine water in the Northwest Pacific Ocean by using statistically processed observational data. We also evaluated the contribution of riverine water to oceanic $CO_2$ uptake and coastal acidification by comparing the TA and DIC values in the sea area affected by riverine water and surrounding sea area.
The Area A affected by riverine water (Area A) was within 250 km of mainland Japan and, to some extent, along the Kuroshio axis. This area was consistent with the low-SSS area on the Pacific coast of Japan. In addition, the range increased to 350 km during the period when the Kuroshio Current meandered south, indicating that the Kuroshio Current path influences the spread of riverine water.

Both nTA and nDIC were higher in Area A compared with the surrounding sea area. The main source of TA was riverine water
in summer, which was balanced by a decrease due to horizontal advection in autumn and winter. The DIC flows were more complex than the TA flows because they were influenced not only by horizontal advection but also by vertical advection and biological activities such as photosynthesis and remineralization. Consequently, the residual term, which integrates these unquantified processes, had a greater influence on the DIC budget than the direct riverine water supply. The contribution of biological activity was suggested to be the primary factor in the residual term by quantification the horizontal and vertical
advection effects. Biological activity showed a maximum in winter and two minima in spring and summer. The annual mean

suggests heterotrophic conditions within the mixed layer in Area A. In any case, because the air-water $CO_2$ flux in near-shore areas is still difficult to estimate, a more thorough quantification of DIC flow in relation to riverine water needs to be considered. Seawater $fCO_2$ in Area A decreased mainly in summer because of the supplied riverine water, which has low-temperature water with high nTA. However, riverine water supply was found to have virtually no effect on oceanic $CO_2$ uptake at an annual

scale. This is because $CO_2$ emission was enhanced by strong wind speeds in winter, in addition to the relatively small increase in seawater $fCO_2$ in winter. The change in the air-sea $CO_2$ flux in winter was coincidentally balanced by the change in summer, and the annual average was almost zero. In Area A, the trend of progressive acidification relative to the surrounding sea area was not confirmed by pH analysis but was confirmed by $\Omega_{arg}$. The supply of TA from riverine water mitigated $\Omega_{arg}$ acidification to 71%. However, an increase in precipitation may have led to higher level of nDIC and acidification in Area A.

This study makes a significant contribution to the analysis of carbon flows in the boundary seawater between terrestrial and oceanic areas because the quantification of carbon flows in these areas is often uncertain in space and time. Enhancing the observational data allows the results to be spatially extended to larger regional or global scales. The analysis results are expected to have a smaller error range owing to the high spatiotemporal resolution of the vertical profiles of the carbonate data and air-water $CO_2$ flux data in near-shore areas.


## Data Availability

The SSS, SST, and $fCO_2$ in air and seawater datasets are available in the SOCAT database (Pfeil et al., 2013; Bakker et al., 2016, https://socat.info/index.php/data-access/). The TA and DIC data were available from the GLODAP database version 2.2020 (Key et al., 2015; Olsen et al., 2016; Olsen, 2020, https://glodap/info/). Wind speed data were obtained from the CCMP

database (version 2.0) (Atlas et al., 2011; Mears et al., 2019; https://data.remss.com/ccmp/v02.0/). The river flow datasets were obtained from the Water Information System of the Ministry of Land, Infrastructure, Transport, and Tourism of Japan (http://www1.river.go.jp/). Vertical water temperature datasets for calculating MLD were obtained from the JCOPE2M dataset (Miyazawa et al., 2017, 2019, . https://www.jamstec.go.jp/jcope/htdocs/distribution/).

## Author Contribution

TT designed the study and wrote the first draft on the manuscript. SN, ST, SS, SD, KE, MI, NK, TO, KT, and YN supplied and managed data for SOCAT and GLODAP. SN supported data processing. All authors contributed to manuscript writing and proofing.

## Competing interests

The contact author has declared that none of the authors have any competing interests.

## Acknowledgments

We appreciate the cooperation of Toyofuji Shipping Co. and Kagoshima Senpaku Co. in the NIES VOS program. We also thank the captains and crews of M/S Pyxis, M/S New Century 2, M/S Trans Future 5, R/V Mirai, R/V Hakuho-maru, R/V Keifu-maru, R/V Ryofu-maru, R/V Wakataka-maru, and R/V Soyo-maru. This research was financially supported by the Global Environmental Research Coordination System, Ministry of the Environment, Japan (grant numbers MOE1751 and MOE 2252) and CREST, Japan Science and Technology Agency (grant number JPMJCR23J4).

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
