# Peer review of "Assessing the impact of riverine water on the Northwest Pacific using normalized Total Alkalinity"

_EGUsphere, 2024_

## Author Response (AR1)

Author's responses

We appreciate the insightful comments from the reviewers. We have responded appropriately to all comments and reflected them in the revised manuscript. Please confirm and review again. The following responses (line xx) correspond to the version without track changes.

Reviewer #1

*GENERAL COMMENTS:*

*This manuscript's primary goal was to analyze the influence of land water on the carbonate systems in the Northwest Pacific region close to Japan using mainly normalized TA (nTA) and DIC (nDIC) with ancillary data of fCO2 and calcite saturation state. The authors also aimed to evaluate the effects of landwater on future climate change, coastal acidification, and environmental changes. Although it is important to continue to further the understanding of the carbonate system dynamics and consequences of climate change and acidification in coastal areas, some aspects of this manuscript are concerning.*

Thank you for reviewing our paper. Below are responses to individual comments.

*First, the authors do not define landwater in the main text, which is the focus of this manuscript. Landwater can be any body of water (e.g., floods, lakes, rivers) on land. The authors show in the supplement material which rivers they selected to represent landwater (Text S2), referencing this information only in the results section of the main text. Furthermore, the authors cite studies of coastal and landwater bodies to generally state that landwaters are responsible for atmospheric CO2 emissions and acidification, which is misleading. Although many studies show that coastal waters can strongly contribute to CO2 emissions, there is still some debate as they can also act as sinks (e.e., Borges et al. GRL, 2005; Mathis et al. Nature Climate Change, 2024). Therefore, I strongly recommend that the authors define the term landwater (generally and for this manuscript) in the main text. I suggest changing to riverine contribution or a similar*

*term, as it is more significant for biogeochemical studies in coastal waters, and it would reach a broader audience.*

We agree that the definition of "landwater" was ambiguous and misleading to the description of air-water CO2 flux. In fact, it was the sum of the river inflow shown in Text S2, so we changed the term to "riverine water" according to the comment. We thought that the definition was obvious, thus we did not add a description of the definition.

*Second, the authors show little knowledge of the carbonate system, as illustrated by some affirmations regarding, for instance, the definition of alkalinity and on what it depends. More concerning, they state that TA could affect air-sea CO2 fluxes, which is mistaken. I suggest revising the chemical concepts and, thus, the consequent discussion and conclusions. Some of the methodology is not well explained and rather referenced to previous work or public datasets. For the purpose of FAIR data, this section of the manuscript should be improved.*

Specifically, we responded to individual comments below, but some misunderstandings are included. The definition of alkalinity proposed by Andrew Dickson is a clear one, but it would be complex and far away from the main content in this study. For example, the explanation about proton donors and acceptors are voluminous but unrelated to the main content in this study. In this paper, we have tried to keep the description as simple as possible, citing a reference for the detailed explanation. Additionally, it is a misunderstanding that alkalinity has no effect on CO2 flux. For example, if DIC is constant and TA increases, the CO2 concentration in seawater will decrease due to the dissociation of CO2 to carbonate ions, and air-water CO2 flux should be affected. Finally, we agree that some parts of the methodology are not fully explained, but we judged that it would be redundant to include the same information as our previous papers (Tokoro et al., 2023) in this paper.

*Third, their methodology strongly relied on the EOF analysis, where their mode decomposition showed that their main three modes explain less than 50% of the spatiotemporal variability (Figure 2 a-c), which questions the degree of confidence of the landwater influence over the study region. Moreover, the discussion does little*

*exploration of other factors that could explain other affecting factors.*

This value of less than 50% is affected by variations north of 37°N. Because of the extreme complexity of the variability in the area, only about half of the variability could be explained by mode 1-3. However, this paper focuses on the variations south of that latitude. Only mode 1 was the dominant explanation for the nTA variation in the area because the EOF analysis results after mode 2 were for variations north of 37°N. Other affecting factors are not listed because it was not possible to assume anything other than riverine water that causes nTA maxima in summer and the effect is inversely proportional to distance from land.

*Finally, the text is not concise and strenuous to follow at times. The introduction section could be better structured. In addition, it lacks important information (references), such as landwater impact in the study region and the magnitude of its contribution to the carbon cycle.*

We improved the text by addressing the individual comments below.

*Based on this consideration and further specific notes, I recommend this article to go under major revision before a second revision by peers.*

*SPECIFIC NOTES:*

*A figure of the studies region (either in the introduction or SM), highlighting the main rivers considered in this study would improve the understanding of the results and the its impact.*

We agree with this comment in part, but the dense distribution of the rivers will make it difficult to see what is illustrated in the figure.

*Lines 32-35: Strong statements. Please define what are is being considered as landwater. Also, please provide a range of how much (ballpark) the land water is contributing to the Co2 emission and acidification.*

As mentioned above, all term of "landwater" was rewritten as "riverine water". We think the definition of this term is obvious. The range of riverine water effects is discussed by the description Figure 5.

*Line 35-36: "However, strong carbon flows, such as biological pumps, impact carbonate distribution in coastal areas". Is it a positive or negative impact?*

It depends on what you consider positive or negative, but we think it affects both sides. For example, increased biological pump in coastal areas will increase atmospheric CO2 absorption, but it will also progress the anoxia on the sediment. We added the description in the main text (Line 36-38).

*Line 39-40: "Therefore, it is crucial to assess the impacts of landwater on oceanic CO2 uptake and coastal acidification". I fail to see why it is important? Is it because of the contrast between the NW Pacific Ocean sink? If so, it would be interesting to provide information (with number) in this paragraph about how much the landwater in Japan contributes to the regional carbon budget.*

What we wanted to indicate is that the assessment of the impact of riverine water to the carbon budget in the Northwestern Pacific, which plays a crucial role in global carbon cycle, is important for understanding the carbon dynamics in this area and predicting future climate change impact. The information about how much the riverine water contributes to the regional carbon budget is not clearly shown in the previous references, and the quantification is the objective of this study. Since the original text was difficult to understand, it has been revised as "The Northwest Pacific Ocean, including the coastal areas of Japan, plays a crucial role in global carbon cycle due to the strong sinks of atmospheric CO2 (Takahashi et al., 2002, 2009)." (Line 42-43).

*Line 43: "variations in landwater in seas in this region". Did you mean just landwater in this region? Explain or rephrase it.*

We mean the contribution of riverine water in this region. The text was revised as "variations in riverine water contribution in this region" (Line 45).

*Line 43: "land water tracers". For the public that do not know, please provide some examples of land water tracers.*

Salinity, stable isotope, and geogenic solutes like silica are representative tracers. We added the information in the text (Line 46)

*Line 45: "other factors such as precipitation and evaporation also impact". Salinity can signal effects of precipitation and evaporation with fresher and saltier values respectively. On this statement, did you mean that precipitation and evaporation have not been intricacy studied such as with the use of isotopes as oxygen-18?*

No. Many studies using salinity and O18 as tracers have quantified the effects of precipitation and evaporation. What we are trying to show here is that salinity alone is not a direct tracer for riverine water, not like nTA.

*Lines 45-47: "This omission ... of and-water ... been observed." (1) remove "and-"; (2) This is an information on effects of precipitation and evaporation on the open ocean. What is the relation with the previous sentence? How is the land water impacting the coastal area of Japan?*

(1) We deleted the word. (2) Although the scope of this study is the coastal area, the analysis includes the open ocean up to 1000 km from land (Fig. 2e) to identify the range of the coastal area. Therefore, using salinity as a tracer may cause error in the identification of the extent.

*Line 50: "defined by the charge balance of dissolved ions, such as hydrogen carbonate (Zeebe and Wolf-Glad-row, 2001)". For general understanding, one can state this definition. However, it is not accurate. TA is defined after Andrew Dickson DSR Part A (1981) as the balance between protons acceptors and proton donors. Please correct the reference and state that this is a broad definition of TA.*

We added "broadly" in the text (Line 53). As mentioned above, the definition of Andrew

Dickson is too long and far from the main contents as the description in this study. We cited Zeebe and Wolf-Gladrow 2001 which contains the definition of Andrew Dickson (Line 54).

*Lines 51-52: "TA depends on several factors, such as advection from different water masses and biological metabolism, including the calcification and dissolution of calcium carbonate." Depends is a strong verb here. TA depends on its sources, which essentially is the process of rock weathering which will provide the major ions, with some biological processes as stated. More currently, the discharge of high nutrient waters due to agriculture has strongly altered TA in some coastal regions. The advection of water masses only displaces the water parcel with a X TA value. Thus, please rephrase this sentence. Also, I suggest looking into these references instead: Zeebe and Wolf-Gladrow (2001); Wolf-Gladrow et al. Mar Them 106 (2007); Kerr et al. Mar Them 237 (2021).*

The word "depends" was replaced to "changes" according to the comment (Line 54). The references you gave us were very helpful in advancing our understanding.

*Line 55: Please add reference for this equation*

The references in the previous sentence (e.g., Broecker and Peng, 1982; Lee et al., 2006; Millero et al., 1998) are applicable. We moved the references to the sentence (Line 57).

*Lines 57-58: " Equation (1) is formulated based on the assumption that a water mass with zero salinity has zero TA." Reference?*

It is obvious that TA = 0 when S = 0 in the equation, otherwise the right-hand side of Equation (1) would go to infinity. We added the explanation (Line 61).

*Line 63: "langwater" landwater?*

We replaced to "riverine water".

*Line 72: "on the environment" It is too general. Please specify on which environment (e.g., the NW Pacific Ocean, Japan's coastal region).*

We rewrote as "environment in riverine water-affected area" (Line 76-77).

*Lines 72-74: "Seawater CO2 fugacity (fCO2) and the calcite saturation state of seawater (Ωcal) were the two carbonate parameters that were used as the index of environmental changes caused by landwater input." Why these two?*

This is because these parameters are related to atmospheric CO2 exchange and ocean acidification, respectively, as explained in the following sentences.

*Line 74: "affects–the" remove the dash*

We corrected the words.

*Lines 74-75: "The TA and DIC supplied by landwater should change seawater fCO2 and oceanic CO2 uptake." TA does not affect the air-sea CO2 flux as CO2 does not constitute TA, hence, only DIC affects air-sea CO2 balance. Please rephrase this sentence.*

This is misunderstanding. TA certainly does not contain CO2, but if TA changes, the CO2 concentration should actually change because the equilibrium among CO2, bicarbonate ion, and carbonate ion should change. For example, if TA increases at the same DIC, CO2 concentration should decrease due to the CO2 dissociation into the ions.

*Line 76: "is an index of ocean acidification". Why not aragonite since it is more sensible to acidification than calcite, specially in open ocean conditions? Please explain in a sentence your choice on using calcite instead of aragonite.*

Calcite was used because foraminifera, coccolithophorid, and bivalves was assumed to

be main species affected by acidification in the study area. Especially, bivalves are important species affected by coastal acidification. We added the explanation in the text (Line 81-82).

*Line 77: "supplied by landwater" Rivers or industrial discharge? What is the definition of landwater used in the manuscript?*

As mentioned above, "landwater" was replaced by "riverine water".

*Line 76-79: "In coastal areas ... Wallace et al., 2014)." This sentence seems better placed at the beginning of the introductions where explaining the importance of studying the impacts of land water.*

We moved the sentence to the first paragraph (Line 38-40) according to the comment.

*Line 96: "temporal data (n > = 60)" Does this correspond to 6 years? Please clarify in the text.*

Yes. This corresponds to 6-years data. The threshold was defined from statistical analysis in the previous study (Line 99).

*Line 110-111: "we identified that "Area A" was significantly influenced by landwater." This seems like a results phrase because I was hoping for a citing figure to show this.*

This sentence only defines the label for "Area A" and does not show the specific extent of the area. We rewrote as "we identified the area significantly influenced by riverine water and labeled as "Area A"" (Line 117-119).

*Line 113: "the value of Area A minus that of Area B" Do you mean, the nTA value of Area A minus the nTA value of area B? Please clarify.*

We defined dAB as the value of Area A minus the value of Area B for all relevant

parameters such as SST, SSS as well as nTA (Line 119-121).

No. The equation 2 has nothing to do with EOF analysis. EOF analysis is used only to identify Area A.

Line 117: "to exclude the effect of seawater volume change due to landwater inflow" Is this a next step? Seems that there is some information missing to support this phrase.

We agree that this sentence is misleading. Here we simply want to indicate that dAB in the equation was calculated from the normalized values, thus the relevant sentence was deleted.

Lines 118-122: "Flux is the term ... to the DIC input" I suggest separating this section in smaller sentences for clarity.'Cflux is the term of difference in air-sea CO2 flux between Areas A and B divided by mixed layer depth (MLD). The MLD was calculated from the reanalysis data of seawater temperature profiles by Japan Agency for 120 Marine-Earth Science and Technology (JCOPE2M; Miyazawa et al., 2017, 2019, https://www.jamstec.go.jp/jcope/htdocs/distribution/). We determined an isothermal depth at ∆T = 0.2 °C with linear interpolation (de Boyer Montégut et al., 2004; Holte and Talley, 2009).'
Also, the MLD is receptive to each area, correct?

We rewrote the sentences according to the comment. The MLD was processed with the same spatiotemporal resolutions with other parameters like nTA, thus is receptive to each area (Line 126-128).

The term of $C_{flux}$ (Line 129-130) because air-sea $CO_2$ flux does not change TA. Please note that this is consistent with TA affecting air-sea $CO_2$ flux as mentioned above.

*Line 124: "for river discharge" First time rivers are mentioned. Is this your definition for landwater?*

Yes. Thus the term landwater was replaced by riverine water.

*Line 131: "Notably" I don't think it is notable to someone unfamiliar to the region. Just remove the word from the sentence. However, DIC has seasonal variations then?*

The word was removed. The seasonal variation of DIC was included in the DIC range (±200 µmol kg-1, Tokoro et al., 2021). The seasonal variation of TA is considered to be negligible from referential data (Line 138-142).

*Lines 131-132: "however, the effect of spatial differences among the three bays was more pronounced." Is there a figure to show this? Please provide one.*

TA ranges between the three bays are listed (518-1006 µmol kg$^{-1}$, Line 1366). Since there was no evidence for seasonal variations of TA, a reference about TA variation of global river was added (Romero-Mujalli et al., 2018, Line 139). The seasonal variation in the literature is a few tens of µmol kg$^{-1}$, which is an order of magnitude smaller than the inner bays range and negligible. We rewrote the description related to the above (Line 139-143).

*Lines 134-137: Generally for the purpose of data transparency, it would be beneficial to have a simple equation showing the general calculation of the CO2 flux (e.g, FCO2 = (fCO2_air-fCO2_sw)\*solubility_product\*transfer_coefficient). The internal calculations of fCO2 by the instrument could be cited to the previous study then, although a mention of which instrument was used also help to determine the whole internal calculations process of fCO2. Please add a general equation for the air-sea CO2 flux.*

We added the description about air-sea CO2 flux according to the comment (Line 144-148).

*Lines 140-144: "Although both seawater fCO2 and Ωcal can be unambiguously determined ... between SSS and nTA and nDIC". Please clarify, were fCO2 and calcite saturation state calculated from a multivariate linear model using TA and DIC? If so, why not use well stablished calculation routines (seacarb, CO2SYS, pyCO2SYS) which consider the non-linearity of the carbonate system parameters?*

No, calcite saturation state was calculated as the same equation of CO2sys. We added the information as ".Ωcal was determined using the equation of a calcite solubility product (Mucci, 1983) and the concentration of Ca2+ and CO32- estimated using the CO2SYS program (Lewis and Wallace, 1998). " (Line 153-155). The multivariate model was used to evaluate the relationship between these parameters and SST, SSS, nTA , and nDIC in a linear and intuitively understandable form.

*Line 147: "was attributed to freshwater inflow from the Amur River in the north and high evaporation at horse latitudes." What was the average inflow of Amur river and levels of evaporation in the horse latitude area compared to other analysed river, areas to make this attribution?*

We rewrote this sentence as "which was attributed to the high-salinity Kuroshio current and the relatively low salinity Oyashio current." because it is more appropriate to use high-salinity Kuroshio and relatively low-salinity Oyashio as a general explanation for the SSS gradient (Line 162-163).

*Line 148: "(32–34°N, 132–140°E; Figure 1a" A rectangle in Figure 1a could help the visualisation.*

We added the rectangle according to the comment.

*Line 151-152: "which was consistent with the TA value of the Amur River ... and Japanese river water" Does this information partially answers my observation for line 147?*

Although the original Line 147 sentence was rewritten, the interception would be the

answer that riverine water is the primary cause of the SSS variation except for Kuroshio-Oyashio gradient (Line 167-169).

*Line 153: "nTA was inversely proportional to SSS" This is an artfact of the calculation of normalisation.*

We agree with the comment. We modified the text for clarity as "Because the intercept value was above zero, nTA seemed to be inversely proportional to SSS (R2 =0.57)". (Line 169-172).

*Lines 165-166: "and maximum nTA was observed in summer south of 37°N, whereas the maximum was in winter north of 37°N (data not shown)" Sentence not clear. What is the second maximum (winter) referred to?*

This meant that the annual maximum was observed in summer and winter at the area of south and north 37N, respectively. We modified the text as "Annual maximum of nTA was observed in summer in the area south and north of 37°N, while it was in winter in the area north of 37°N (data not shown)." (Line 183-184).

*Line 166: "thereby reaching its peak in summer on the Pacific side" Is there precipitation data to support this results or is it based on the literature/meteo data center? If so, please provide a reference.*

It is based on the database of Japan Meteorological Agency. We added the information (Line 186).

*Lines 197-198: "of all dAB tended to increase significantly" even SST (Figure 3e)?*

Yes. We checked the p-value of the linear regression for all parameters including SST.

*Line 198: "This indicates that the supply of landwater increased" One removes the influence of riverine input when applying the traditional equation to normalize TA. What*

*does measured TA show?*

We already discussed in the introduction that the traditional equation could not remove the influence of riverine water above TA zero. Rather, nTA is a quantitative tracer of the impact of riverine water, so the temporal change of riverine water effect can be assessed using nTA.

Reviewer #2

*I have seen the comments by the other reviewer and mostly agree with them. They do make a comment about TA which I do not think is correct. Specifically their comment on lines 74-75 is incorrect. Changes in TA (and DIC) do indeed affect seawater $pCO_2$ and therefore air-sea $CO_2$ fluxes, as the authors state. Perhaps the confusion arises because the opposite is indeed not true (air-sea $CO_2$ fluxes do not affect TA).*

We agree the definition of TA that TA is determined by the difference in concentration between the proton donor and acceptor is accurate. However, the detailed explanation of proton donor and other factors are not very relevant to the main contents and are lengthy, thus the description has been changed to refer to the cited reference (Zeebe and Wolf-Gladrow, 2001) (Lines 54).
As for TA and CO2 flux, we agree with you that it is a misunderstanding.

*I am not very familiar with EOF analysis and I find that the explanation in the text is very brief and does not really help understand how to interpret the results. I think that a short paragraph with a conceptual explanation is needed, especially because this is not the usual way to determine these processes (where usually one would instead do direct calculations from the measured carbonate system parameters). The lack of this explanation makes it difficult for me to assess the authors' attribution of the different processes to the various signals seen in the data. Of course it's not necessarily a negative that the usual methods have not been used here because there should be scope to try different ways, maybe this can deliver some different insight. But that is not clear in this version of the manuscript.*

The EOF analysis can quantify multiple dominant fluctuations from the overall

fluctuation. The target area in this study is expected to be affected by multiple fluctuations such as seasonal changes of the Kuroshio and Oyashio flows, in addition to the effect of riverine water (note that all word "landwater" was replaced with "riverine water" according to the other reviewer's comment). Therefore, it is easier to extract the influence of riverine water using the EOF analysis than using conventional methods with direct use of carbonate parameters. We added the description about the above explanation in the text (Line 113-117).

*The point mentioned by the other reviewer about the first three EOF modes capturing less than 50% of the total variability seems to be a serious issue that does need to be resolved.*

As shown in the responses to other reviewer below, most of the remaining 50% contributes to variation north of 37°N, which is outside the target in this study. "This value of less than 50% is affected by variations north of 37°N. Because of the extreme complexity of the variability in the area, only about half of the variability could be explained by mode 1-3. However, this paper focuses on the variations south of that latitude. Only mode 1 was the dominant explanation for the nTA variation in the area because the EOF analysis results after mode 2 were for variations north of 37°N."

*44-45    Precipitation and evaporation affect both salinity and total alkalinity and in a perfectly proportional way – so it's not clear why these processes would cause problems here?*

The problem is that the effect of riverine water, which is not in a proportional with salinity, cannot be quantified when only salinity or TA, when it is affected by precipitation and evaporation. You are correct that these parameters variations due to precipitation and evaporation are perfectly proportional, thus nTA can be used to remove the effects of precipitation and evaporation.

*50            TA is not based on charge balance. Some of the terms do have coefficients that match their charges, but not all, and some uncharged species appear in the equation. Instead, TA is defined based on the capacity to accept or donate protons (H+). Please update accordingly.*

As mentioned above, a strict definition of TA could be redundant, so we stated to refer to the citation in the text (Line 54).

*53          Not clear why there is a "however" here.*

We deleted the word according to the comment.

*64          "langwater" => "landwater"*

All word "landwater" was replaced by "riverine water" according to the comment of other reviewer's comment.

*112          Please refer to the map of areas A and B here.*

The definitions of the labels for areas A and B are described here, and the actual extent of areas A and B has not yer been determined. Therefore, referring to the map would be confusing.

*118          Presumably Cflux is zero for TA?*

Yes, thus the term was applied only to DIC (Line 129-130).

*140-143          As the authors note, fCO2 and Ω can be calculated directly from TA and DIC. Then the drivers of their changes can be exactly understood – there are many studies that do this both from a perspective of laying out the theory, explaining it conceptually so it can be understood intuitively, and applying it to real datasets.*

We may not fully understand this comment, but we would like to argue here that even if fCO2 and Ω can be determined in a non-linear way, re-presenting the relationship in a linear way is more useful for intuitive understanding, for example by means of a graphical representation such as Figure 5.

*141-142          It doesn't make sense to me that a multivariate linear model would be useful **because** there are NON-linear relationships?*

We do not agree with this comment. For example, when an explanatory variable like nTA is changed, it is easy to predict how much fCO2 and Ω will be affected in a multivariate linear model. In a nonlinear model such quantitative prediction by the change of the explanatory variables is difficult.

*Figure 2          The distribution of areas A and B does not look very natural*

*(coarse boundary with many right angles and straight sections). Is this really the best way to define the region of landwater influence? Also, looking at panels a-c, I cannot see why these regions were selected – they don't seem to correspond to any features in a-c, although the text implies that a-c were used to choose the regions. It is mentioned (235) that area A is consistent with a low-salinity regime; might it make more sense to define the areas in terms of salinity, for example?*

The distribution of areas A and B appears unnatural due to the spatial resolution of $1° \times 1°$ in this study. With this resolution, it is impossible to create a distribution that looks any more natural.

The panels a-c do not provide a direct definition of areas A and B. These panels indicate that the area south of $37°$ is the area to be analyzed in this study. Area A is defined as the area determined from the distance dependence of the temporal variation of Mode 1 south of $37°$ (panel d, e). Area B is determined from the definition in the text (Line 119).

It is pointed out in the introduction (Line 47-49) that the definition of area A using salinity would be erroneous due to the effect of precipitation and evaporation.

*360-362                              Specifying these distances to two decimal places seems unrealistic – to the nearest (ten) kilometre(s) would seem to make more sense.*

We agree with this comment. The distance values were replaced by rounded values to two digits (ten kilometers).

---

## Referee Report (RR1)

The authors aim to analyze the influence of riverine input on the $CO_2$ system in the northwestern Pacific, and its implications for future climate change and coastal acidification. Their analysis focuses on four key parameters of the $CO_2$ system: total alkalinity (TA), dissolved inorganic carbon (DIC), calcite saturation state ($\Omega$cal), and fugacity of $CO_2$ (f$CO_2$).

The manuscript presents an interesting and relevant assessment of the effects of freshwater discharge from the eastern coast of Japan into the northwestern Pacific Ocean. The statistical approach is sound, and the manuscript is well-organized. It should be considered for publication, provided the discussion of acidification processes is further improved.

In the Introduction, the authors state that riverine input is the major carbon source for the oceans. However, this claim is not supported by current estimates. Rivers discharge approximately 0.9 to 1.3 Pg of carbon per year into the ocean, in both dissolved and particulate forms. A recent global assessment by Liu et al. (2024) estimates riverine carbon export at $1.02 \pm 0.22$ ($2\sigma$) Pg C yr$^{-1}$, partitioned into $0.52 \pm 0.17$ Pg C yr$^{-1}$ of dissolved inorganic carbon, $0.30 \pm 0.14$ Pg C yr$^{-1}$ of dissolved organic carbon, $0.18 \pm 0.04$ Pg C yr$^{-1}$ of particulate organic carbon, and $0.03 \pm 0.02$ Pg C yr$^{-1}$ of particulate inorganic carbon. In contrast, the ocean's uptake of atmospheric $CO_2$ is estimated to be significantly higher—around $2.9 \pm 0.4$ Pg C yr$^{-1}$ (Friedlingstein et al., 2025). Therefore, on a global scale, atmospheric carbon uptake is a more substantial source of carbon to the ocean than riverine input. It should be noted, however, that earlier estimates, such as those by Bauer et al. (2013), suggested that in coastal regions specifically, riverine carbon inputs could exceed atmospheric contributions. This distinction between global ocean and coastal zone carbon budgets should be clarified in the manuscript.

The abstract addresses the issue of coastal acidification; however, this aspect is only indirectly considered in the manuscript, as the selected parameters do not include pH. Instead, the authors use the calcite saturation state ($\Omega$cal) as a proxy for acidification. While $\Omega$ is commonly used in ocean acidification (OA) studies, it is important to note that in the global surface ocean, pH and $\Omega$ often exhibit out-of-phase behavior in terms of both spatial distribution and seasonal variability (e.g., Xue et al., 2021; Kwiatkowski & Orr, 2018), which may seem counterintuitive. Although both pH and $\Omega$ are widely used as indicators of OA, their asynchronous variability complicates the choice of which parameter better represents the impact of carbonate chemistry on marine organisms (Jokiel, 2013; Waldbusser et al., 2014). This complexity arises from the interplay between long-term trends and short-term natural variability, which can obscure biological responses (Kwiatkowski & Orr, 2018; Landschützer et al., 2018).

Many of the biological effects of ocean acidification depend not only on carbonate ion concentration [$CO_3^{2-}$] but also on hydrogen ion concentration [H+], i.e., pH, as well as on bicarbonate [$HCO_3^-$] (Cyronak et al., 2015; Jokiel, 2013). If the authors intend to explore the impacts of acidification in depth, it is essential that they include and discuss pH data alongside the other $CO_2$ system parameters. Without pH, the discussion of acidification remains incomplete and indirect. Alternatively, if the focus is primarily on the carbonate saturation state, the manuscript should clearly state this and align the discussion accordingly.

Furthermore, the authors report $\Omega$ (omega) for calcite but do not include $\Omega$ for aragonite, which is more commonly used in ocean acidification studies. Aragonite is more soluble than calcite, and organisms that produce aragonitic shells or skeletons are generally more vulnerable to decreasing saturation states under acidified conditions. Therefore, aragonite saturation is typically the preferred indicator when assessing biological sensitivity to ocean acidification. The manuscript would benefit from a clearer justification for the choice of $\Omega$_calcite over $\Omega$_aragonite, or, from the inclusion of both parameters to allow for a more comprehensive assessment.

**Minor comments**

L.21 The sentence should be revised: the affirmation that the riverine wate is not the dominant cause of Dissolved Inorganic Carbon should be better formulated.

L. 23 The supply of by riverine water ": correct.

L371-374 and 380 and 398. The sentences referring to acidification should be revised for clarity and accuracy. As the authors do not include pH data in their analysis, they are not directly addressing the acidification process, which is fundamentally defined by changes in hydrogen ion concentration [H+]. Instead, their focus is on the carbonate saturation state ($\Omega$), which is indeed affected by acidification but also depends on the buffering capacity of the system.

Given this, I suggest that the authors frame their discussion primarily in terms of the saturation state of the studied system, rather than acidification per se, unless they choose to explicitly include calculated or measured pH values. Including pH—either measured or derived from the available $CO_2$ system parameters—would allow for a more comprehensive and direct discussion of the acidification process.

L. 386-387 Regarding the statement that the DIC flow were more complex than the TA flow it is not very clear if the authors refer to the advection to the biologically related process, they should explain better how the flow is affected by a residual term -- which are the processes behind it?

L174-176. "Please add units to the colour bars in panels (b) and (c) of Figure 1 to ensure clarity and consistency. Additionally, the significance level (e.g., *p*-value) for the linear regression should be provided to assess the statistical robustness of the trend. The term 'approximate line' is ambiguous—please clarify its meaning. If it refers to a fitted trend line, specify the method used (e.g., least squares regression).

L. 237-240 in figure 4 b and e I suppose that the units should be $\mu mol\ kg^{-1}\ month^{-1}$ not $year^{-1}$

L280-281 However, no regional difference in calcification rate of up to 10 $\mu mol\ kg^{-1}\ year^{-1}$ has not been reported... The are two negations which means an affirmation The sentence should be corrected.

L. 331 In figure 6a the units should be $\mu mol\ kg^{-1}\ month^{-1}$ not $year^{-1}$. As the data represented are monthly averages.

L.345-347 Due to the wide offshore oceanic area with respect to the shallow coastal waters of I wonder if the role of submerged aquatic vegetation can be considered relevant. Could you provide more support to this affirmation.

L.369 Correct "air-sea CO2 flues".

**Text S1 Vertical profile an advection**

"They ranged frm" : correct

**Text S2 Riverine supply from mainland of Japan**

It is not clear to me if the authors assumed the same flow rate for all the rivers or if the compute the total riverine flow considering a proportionality with the flow rate of each river. Could you explain better this part.

Could the authors provide a table similar to S1 comparing the characteristic of area A and B in order to show the different oceanographic characteristics

"The English throughout the manuscript should be revised by a native speaker or a professional language editor to improve clarity, and overall readability.

**References**

Bauer, J. E., Cai, W. J., Raymond, P. A., Bianchi, T. S., Hopkinson, C. S., and Regnier, P. A. G.: The changing carbon cycle of the coastal ocean, Nature, 504, 61–70. https://doi.org/10.1038/nature12857, 2013.

Cyronak, T., Schulz,K.G., Jokiel,P.L. 2015.TheOmega myth: what really drives lower calcification rates in an acidifying ocean. ICES Journal of Marine Science, doi: 10.1093/icesjms/fsv075.

Friedlingstein, P. et al. 2025. Global Carbon Budget 2024 Earth Syst. Sci. Data, 17, 965–1039.

Jokiel, P. L. 2013. Coral reef calcification: Carbonate, bicarbonate and proton flux under conditions of increasing ocean acidification. Proceedings of the Royal Society B: Biological Sciences, 280(1764), 529–549. https://doi.org/10.1098/rspb.2013.0031

Kwiatkowski, L., & Orr, J. C. (2018). Diverging seasonal extremes for ocean acidification during the twenty-first century. Nature Climate Change, 8(2), 141–145. https://doi.org/10.1038/s41558-017-0054-0

Landschützer, P., Gruber, N., Bakker, D. C. E., Stemmler, I., & Six, K. D. (2018). Strengthening seasonal marine CO2 variations due to increasing atmospheric CO2. Nature Climate Change, 8(2), 146–150. https://doi.org/10.1038/s41558-017-0057-x

Liu, M., Raymond, P.A., Lauerwald, R. *et al*. Global riverine land-to-ocean carbon export constrained by observations and multi-model assessment. *Nat. Geosci*. **17**, 896–904 (2024). https://doi.org/10.1038/s41561-024-01524-z)

Xue, L., Cai, W.-J., Jiang, L.-Q., Wei, Q. 2021. Why are surface ocean pH and CaCO3 saturation state often out of phase in spatial patterns and seasonal cycles? Global BiogeochemicalCycles, 35, e2021GB006949. https://doi.org/10.1029/2021GB00694

Waldbusser, G. G., Hales, B., Langdon, C. J., Haley, B. A., Schrader, P., Brunner, E. L., et al. 2014. Saturation-state sensitivity of marine bivalve larvae to ocean acidification. Nature Climate Change, 5, 273–280. https://doi.org/10.1038/NCLIMATE2479

---

## Author Response (AR2)

**Responses to reviewer #2**

*The authors aim to analyze the influence of riverine input on the $CO_2$ system in the northwestern Pacific, and its implications for future climate change and coastal acidification. Their analysis focuses on four key parameters of the $CO_2$ system: total alkalinity (TA), dissolved inorganic carbon (DIC), calcite saturation state ($\Omega cal$), and fugacity of $CO_2$ ($fCO_2$).*

*The manuscript presents an interesting and relevant assessment of the effects of freshwater discharge from the eastern coast of Japan into the northwestern Pacific Ocean. The statistical approach is sound, and the manuscript is well-organized. It should be considered for publication, provided the discussion of acidification processes is further improved.*

Thank you for your very helpful comments and references. These were extremely valuable for brushing up our paper. Below are our responses to each comment. We would appreciate it if you could confirm them.

*In the Introduction, the authors state that riverine input is the major carbon source for the oceans. However, this claim is not supported by current estimates. Rivers discharge approximately 0.9 to 1.3 Pg of carbon per year into the ocean, in both dissolved and particulate forms. A recent global assessment by Liu et al. (2024) estimates riverine carbon export at $1.02 \pm 0.22$ ($2\sigma$) Pg C yr$^{-1}$, partitioned into $0.52 \pm 0.17$ Pg C yr$^{-1}$ of dissolved inorganic carbon, $0.30 \pm 0.14$ Pg C yr$^{-1}$ of dissolved organic carbon, $0.18 \pm 0.04$ Pg C yr$^{-1}$ of particulate organic carbon, and $0.03 \pm 0.02$ Pg C yr$^{-1}$ of particulate inorganic carbon. In contrast, the ocean's uptake of atmospheric $CO_2$ is estimated to be significantly higher— around $2.9 \pm 0.4$ Pg C yr$^{-1}$ (Friedlingstein et al., 2025). Therefore, on a global scale, atmospheric carbon uptake is a more substantial source of carbon to the ocean than riverine input. It should be noted, however, that earlier estimates, such as those by Bauer et al. (2013), suggested that in coastal regions specifically, riverine carbon inputs could exceed atmospheric contributions. This distinction between global ocean and coastal zone carbon budgets should be clarified in the manuscript.*

We agree with the numerical comparison above. In this introduction, we intended to show that the flux from riverine water is of the same order of magnitude and importance as the flux from the atmosphere. We have rewritten the text to make the quantitative comparison clearer (Line 33-39).

*The abstract addresses the issue of coastal acidification; however, this aspect is only indirectly considered in the manuscript, as the selected parameters do not include pH.*

*Instead, the authors use the calcite saturation state ($\Omega$cal) as a proxy for acidification. While $\Omega$ is commonly used in ocean acidification (OA) studies, it is important to note that in the global surface ocean, pH and $\Omega$ often exhibit out-of-phase behavior in terms of both spatial distribution and seasonal variability (e.g., Xue et al., 2021; Kwiatkowski & Orr, 2018), which may seem counterintuitive. Although both pH and $\Omega$ are widely used as indicators of OA, their asynchronous variability complicates the choice of which parameter better represents the impact of carbonate chemistry on marine organisms (Jokiel, 2013; Waldbusser et al., 2014). This complexity arises from the interplay between long-term trends and short-term natural variability, which can obscure biological responses (Kwiatkowski & Orr, 2018; Landschützer et al., 2018).*

*Many of the biological effects of ocean acidification depend not only on carbonate ion concentration $[CO_3^{2-}]$ but also on hydrogen ion concentration $[H^+]$, i.e., pH, as well as on bicarbonate $[HCO_3^-]$ (Cyronak et al., 2015; Jokiel, 2013). If the authors intend to explore the impacts of acidification in depth, it is essential that they include and discuss pH data alongside the other $CO_2$ system parameters. Without pH, the discussion of acidification remains incomplete and indirect. Alternatively, if the focus is primarily on the carbonate saturation state, the manuscript should clearly state this and align the discussion accordingly.*

*Furthermore, the authors report $\Omega$ (omega) for calcite but do not include $\Omega$ for aragonite, which is more commonly used in ocean acidification studies. Aragonite is more soluble than calcite, and organisms that produce aragonitic shells or skeletons are generally more vulnerable to decreasing saturation states under acidified conditions. Therefore, aragonite saturation is typically the preferred indicator when assessing biological sensitivity to ocean acidification. The manuscript would benefit from a clearer justification for the choice of $\Omega$_calcite over $\Omega$_aragonite, or, from the inclusion of both parameters to allow for a more comprehensive assessment.*

We agree that omitting a description of pH is very problematic for analyzing the discussion of acidification. The distinct spatiotemporal behavior of pH and $\Omega$, as indicated in the references in this comment, is highly intriguing. Therefore, we have added a description of the analysis of pH together with $\Omega$, which has been recalculated as an aragonite-based one according to the comment. Interestingly, a clear acidification trend was not confirmed by the pH results (see Figure 5). This was due to riverine water causing low SST in Area A, particularly in winter, which counteracted the effects of nDIC inflow. The revised manuscript therefore states that, while riverine water inflow has little effect on hydrogen ion concentration, it does influence biological activities such as calcification (Line 399-410, 434-437). Furthermore, while the effects of hydrogen ion concentration and various carbonate

ion concentrations on biological activity are interesting, we found it challenging to provide an individual assessment of these parameters to our manuscript because riverine water inflow affect multiple parameters including SST, SSS, nTA and nDIC, and thus hydrogen ion and carbonate ions simultaneously.

*Minor comments*

*L.21 The sentence should be revised: the affirmation that the riverine wate is not the dominant cause of Dissolved Inorganic Carbon should be better formulated.*
We have rewritten as "The analysis showed that riverine water was the main cause of the higher total Alkalinity compared to the surrounding area, whereas its contribution to the increase in Dissolved Inorganic Carbon was relatively minor." (Line 22-24).

*L. 23 The supply of by riverine water ": correct.*
It was a mistake. We have corrected it.

*L371-374 and 380 and 398. The sentences referring to acidification should be revised for clarity and accuracy. As the authors do not include pH data in their analysis, they are not directly addressing the acidification process, which is fundamentally defined by changes in hydrogen ion concentration [H+]. Instead, their focus is on the carbonate saturation state ($\Omega$), which is indeed affected by acidification but also depends on the buffering capacity of the system.*
*Given this, I suggest that the authors frame their discussion primarily in terms of the saturation state of the studied system, rather than acidification per se, unless they choose to explicitly include calculated or measured pH values. Including pH—either measured or derived from the available $CO_2$ system parameters—would allow for a more comprehensive and direct discussion of the acidification process.*
As responded to above major comment, the descriptions of pH have been added. As a result, differing behavior was observed: while pH showed little change, omega exhibited a clear acidification trend. As noted in the comment, including pH enabled a comprehensive analysis of acidification (Figure 5, Line 399-4104, 34-437, and numerous other lines).

*L. 386-387 Regarding the statement that the DIC flow were more complex than the TA flow it is not very clear if the authors refer to the advection to the biologically related process, they should explain better how the flow is affected by a residual term -- which are the processes behind it?*

We have rewritten the sentence as "The DIC flows were more complex than the TA flows because they were influenced not only by horizontal advection but also by vertical advection and biological activities such as photosynthesis and remineralization." (Line 421-423).

*L174-176. "Please add units to the colour bars in panels (b) and (c) of Figure 1 to ensure clarity and consistency. Additionally, the significance level (e.g., p-value) for the linear regression should be provided to assess the statistical robustness of the trend. The term 'approximate line' is ambiguous—please clarify its meaning. If it refers to a fitted trend line, specify the method used (e.g., least squares regression).*

We have corrected the figure and caption accordingly (Figure 1, Line 185-188). The approximate line is the linear regression line using the least squares method.

*L. 237-240 in figure 4 b and e I suppose that the units should be $\mu$ mol kg-1 month-1 not year-1*

The unit have changed to be per month accordingly (Figure 4).

*L280-281 However, no regional difference in calcification rate of up to 10 $\mu$ mol kg-1 year-1 has not been reported⋯ The are two negations which means an affirmation The sentence should be corrected.*

It was a mistake. We have corrected it (Line298-299).

*L. 331 In figure 6a the units should be $\mu$ mol kg-1 month-1 not year-1. As the data represented are monthly averages.*

We have corrected it like Figure 4 (Figure 6).

*L.345-347 Due to the wide offshore oceanic area with respect to the shallow coastal waters of I wonder if the role of submerged aquatic vegetation can be considered relevant. Could you provide more support to this affirmation.*

The words "submerged aquatic vegetation" described on the image of coastal observation, but there is no data guaranteeing it has a significant impact on the study area. Therefore, the words have been removed (Line 373).

*L.369 Correct "air-sea CO2 flues".*

A mis typo. We have corrected it (Line 396).

*Text S1 Vertical profile an advection*

*"They ranged frm" : correct*
A mis typo. We have corrected it.

*Text S2 Riverine supply from mainland of Japan*
*It is not clear to me if the authors assumed the same flow rate for all the rivers or if the compute the total riverine flow considering a proportionality with the flow rate of each river. Could you explain better this part.*
We calculated the flow rate considering the proportion. We agree that Text S2 was generally difficult to understand, so we have completely rewritten it.

*Could the authors provide a table similar to S1 comparing the characteristic of area A and B in order to show the different oceanographic characteristics*
We agree that such table is useful. We added a Table (Table S2, former Table S2 was renamed S3).

*"The English throughout the manuscript should be revised by a native speaker or a professional language editor to improve clarity, and overall readability.*
Thank you for your comment regarding the English of the manuscript. The manuscript had already been proofread by a professional language editing service. In response to the reviewer's suggestion, we have carefully re-checked the text and made additional minor revisions to further improve clarity and readability.

[revised manuscript text omitted]

The total Alkalinity (TA) normalized by salinity is a potential indicator to assess the influence of riverine water on ocean. TA  is defined broadly as the charge difference between  proton donor and acceptor (see Zeebe and Wolf-Gladrow, 2001 for more detailed definition). TA changes by several factors, such as advection from different water masses and biological metabolism, including [26] calcification and dissolution of calcium carbonate. [27] Because TA is also highly correlated with salinity, TA normalized to [28] reference salinity (nTA) has been used to quantify the above [29] factors (e.g., Broecker and Peng, 1982; Lee et al., 2006; Millero et al., 1998). nTA is calculated as follows:

$$\text{nTA} = \text{TA} \cdot \frac{S_{ref}}{S} \tag{1}$$

where, $S$ and $S_{ref}$ are the measured and reference salinities (traditionally 35), respectively. Similarly, Dissolved Inorganic Carbon concentration (DIC) [30] is also normalized (nDIC). Equation (1) is formulated based on the assumption that a water mass with zero salinity has zero TA, otherwise the right-hand side would go to infinity. However, this assumption is not true for riverine water because its TA is greater than zero, even when salinity is zero, owing to the weathering of carbonate and silicate rocks (e.g., Friis et al., 2003; Lehmann et al., 2023; Rassmann et al., 2016; Taguchi et al., 2009). Therefore, when

番号 : 26 作成者 : Frédéric Gazeau   日付 : 2025/09/22 17:23:18

番号 : 27 作成者 : Frédéric Gazeau   日付 : 2025/09/22 17:23:42
a

番号 : 28 作成者 : Frédéric Gazeau   日付 : 2025/09/22 17:23:27

番号 : 29 作成者 : Frédéric Gazeau   日付 : 2025/09/22 17:23:53
mentionned

番号 : 30 作成者 : Frédéric Gazeau   日付 : 2025/09/30 14:41:12
, which is the sum of carbon species,

Equation (1) is applied to areas affected by riverine water, the nTA value is higher than that of the surrounding seawater. Conversely, the influence of riverine water, along with factors such as water mass advection and biological re___sm, can be quantified by assessing the distribution of nTA. Unlike other methods that use salinity as a tracer for riverine water, nTA defined in Equation (1) excludes the effect of precipitation and evaporation, which is advantageous because contamination by water masses with almost zero salinity and TA can be excluded. ___ also easier to quantify the influence of seawater from different local areas because the TA has different values in different local areas, even if the salinity is the same (Lee et al., 2006; Takatani et al., 2014).

In this study, we aimed to analyse the influence of riverine water on  using nTA and other carbonate parameters measured by voluntary cargo ships and research vessels . First, spatiotemporal variations in the area in which riverine water significantly affected surface seawate___e identified using ___irical Orthogonal Function (EOF) analysis of the nTA distribution. Second, we focused on the differences in TA and DIC between the riverine water-affected area and surroundin___as. We quantified the riverine water supply and other contributing factors that affect TA and DIC. The final step involved the evaluation of the effects of riverine water on the environment in riverine water-affected area. Seawater $CO_2$ fugacity ($fCO_2$) and the calcite saturation state of seawater ($\Omega_{cal}$) were the two carbonate parameters that were used as the index of environmental changes caused by riverine water input.  e TA and DIC supplied by riverine water ___ change seawater $fCO_2$ and oceanic $CO_2$ uptake. Meanwhile, ___s the ratio of the concentration product of $[Ca^{2+}]$ and $[CO_3^{2-}]$ to the solubility product of calcite and is an index of ocean acidification. Calcite is a mineral composed of $CaCO_3$ and constitutes foraminifera, coccolithophorids, and ___ er of bivalves. Therefore, the analysis of changing $\Omega_{cal}$ is expected to lead to a more detailed assessment of coastal acidification in the study area. This study also aimed to evaluate the effects of riverine water on future climate change and coastal acidification and to predict the effects of future environmental changes.

**2. Methods**

**2.1 Data for analysis**

The observational data in this study were collected by the National Institute for Environmental Studies (NIES), Meteorological Research Institute (MRI) of the Japan Meteorological Agency (JMA), and Japan Fisheries Research and Education Agency (FREA). The NIES data were produced as part of the Voluntary Observing Ship (VOS) programs for cargo ships (namely, M/S Alligator Hope, M/S Pyxis, M/S New Century 2, and M/S Trans Future 5). MRI, JMA, and FREA collected data from research vessels (R/V Mirai and R/V Hakuho-maru for MRI; R/V Keifu-maru and R/V Ryofu-maru for JMA; and R/V Wakataka-maru and R/V Soyo-maru for FREA). These data were uploaded to the Surface Ocean $CO_2$ Atlas (SOCAT; Pfeil et al., 2013; Bakker et al., 2016, https://socat.info/index.php/data-access/) and Global Ocean Data Analysis Project (GLODAP, Key et al., 2015; Olsen et al., 2016; Olsen et al., 2020, https://glodap.info/). These observations were

番号 : 1 作成者 : Frédéric Gazeau  日付 : 2025/09/22 17:57:11
activity

番号 : 2 作成者 : Frédéric Gazeau  日付 : 2025/09/22 17:57:03

番号 : 3 作成者 : Frédéric Gazeau  日付 : 2025/09/22 17:57:45
I do not understand this sentence, please rephrase

作成者 : Tatsuki Tokoro  タイトル : ノート注釈  日付 : 2025/10/10 15:51:17
We have rewritten as "Furthermore, nTA can be used to quantify the influence of seawater inflow from different local areas because distinct regional differences in nTA have been reported (Kakehi et al., 2017; Lee et al., 2006; Takatani et al., 2014).".(Line 72-74)

番号 : 4 作成者 : Frédéric Gazeau  日付 : 2025/09/22 17:58:36
the crbonate chemistry of the Northwest Pacific

番号 : 5 作成者 : Frédéric Gazeau  日付 : 2025/09/22 17:58:30

番号 : 6 作成者 : Frédéric Gazeau  日付 : 2025/09/22 17:59:05

番号 : 7 作成者 : Frédéric Gazeau  日付 : 2025/09/22 17:59:48
an

番号 : 8 作成者 : Frédéric Gazeau  日付 : 2025/09/22 17:59:30
carbonate chemistry

番号 : 9 作成者 : Frédéric Gazeau  日付 : 2025/10/08 14:52:45
unaffected

番号 : 10 作成者 : Frédéric Gazeau  日付 : 2025/09/22 18:02:43

番号 : 11 作成者 : Frédéric Gazeau  日付 : 2025/10/08 14:55:18
influence

番号 : 12 作成者 : Frédéric Gazeau  日付 : 2025/09/22 18:01:08

番号 : 13 作成者 : Frédéric Gazeau  日付 : 2025/09/22 18:01:52

番号 : 14 作成者 : Frédéric Gazeau  日付 : 2025/09/22 18:03:00
Please check comment from R#2, I concur, you should restrict here the use of omega not from an ocean acidification perspective but changing saturation conditions impacting calcifiers.

作成者 : Tatsuki Tokoro  タイトル : ノート注釈  日付 : 2025/10/10 15:51:34
We agree that omitting pH in the acidification discussion is problematic. However, unlike omega, the calculated pH did not confirm a trend of acidification. This is considered due to the SST dependency of pH. Because the difference between the two acidification index is interesting,  we have decided to describe both pH and omega in the revised manuscript. Furthermore, according to the comment, omega has been changed to the aragonite based one.(Figure 5, Line 399-4104, 34-437, and numerous other lines).

番号 : 15 作成者 : Frédéric Gazeau  日付 : 2025/10/08 14:58:38
shells

番号 : 16 作成者 : Frédéric Gazeau  日付 : 2025/09/22 18:04:56

[revised manuscript text omitted]

番号 : 1 作成者 : Frédéric Gazeau  日付 : 2025/10/08 15:13:12

concentrations

番号 : 2 作成者 : Frédéric Gazeau  日付 : 2025/09/22 18:41:59

concentration in

番号 : 3 作成者 : Frédéric Gazeau  日付 : 2025/09/22 18:42:21

番号 : 4 作成者 : Frédéric Gazeau  日付 : 2025/09/22 18:43:16

Not clear, could you rephrase?

> 作成者 : Tatsuki Tokoro  タイトル : ノート注釈  日付 : 2025/10/10 15:51:47
> We have rewritten as "The DIC range was determined based on the maximum (1171 µmol kg$_{-1}$ in Tokyo Bay) and minimum (675 µmol kg$_{-1}$ in Ise Bay) freshwater endmember values among the three bays, with an additional ±200 µmol kg$_{-1}$ to account for seasonal variation estimated in the previous study (Tokoro et al., 2021)." for clarification. (Line 143-145)

番号 : 5 作成者 : Frédéric Gazeau  日付 : 2025/09/22 18:44:22

it

番号 : 6 作成者 : Frédéric Gazeau  日付 : 2025/09/22 18:44:14

番号 : 7 作成者 : Frédéric Gazeau  日付 : 2025/09/22 18:43:56

what does that mean?

do you mean "based on values from the literature"?

> 作成者 : Tatsuki Tokoro  タイトル : ノート注釈  日付 : 2025/10/10 15:52:16
> The value from a literature means the global range of riverine TA variation due to water temperature change. We have rewritten the sentences for clarification. (Line 145-148)

番号 : 8 作成者 : Frédéric Gazeau  日付 : 2025/09/30 17:04:04

at 10 m

番号 : 9 作成者 : Frédéric Gazeau  日付 : 2025/09/22 18:48:29

Comment from R#2

> 作成者 : Tatsuki Tokoro  タイトル : ノート注釈  日付 : 2025/10/10 14:22:37
> As mentioned above, we have decided to use both pH and omega for the index of acidification.

**3. Results**

Figure 1 represents the processed spatial distributions of SSS, TA, and nTA. The SSS showed a north-south gradient, which was attributed to the high-salinity Kuroshio current and the relatively low salinity Oyashio current. There was also an area of reduced SSS along the Pacific coast of mainland Japan (32–34°N, 132–140°E; Figure 1a). These trends were similar for TA,

165   which exhibited high correlation with SSS ($R^2 = 0.94$) (Figure 1b, d). The nTA was high in the northern part of the study area, and slightly high values were also observed along the Pacific coast of Japan (Figure 1c). The intercept of the regression line between SSS and TA was 528.04 ± 2.00 μmol kg$^{-1}$ (Ave ± SE), which was consistent with the TA value of large river on the continental side like the Amur River (589 μmol kg$^{-1}$; Andreev and Pavlova, 2009) and Japanese river water (518–1006 μmol kg$^{-1}$) (Figure 1d). Because the intercept value was above zero, nTA seemed to be inversely proportional to SSS ($R^2$

170   =0.57), with a lower SSS tending to have a higher nTA (Figure 1e). These results indicate that the study area was affected by freshwater with TA above zero, especially in the northern area and on the Pacific coast of Japan, and that nTA could be used as a tracer for the freshwater.

番号 : 1 作成者 : Frédéric Gazeau 日付 : 2025/09/30 17:23:57
inputs

番号 : 2 作成者 : Frédéric Gazeau 日付 : 2025/09/22 18:56:41

re 1 (a–c): Spatial distribution of (a) mean SSS (b) TA and (c) n□□□he rectangular in (a) show the reduced SSS area along the Pacific coast of mainland Japan (32–34°N, 132–140°E) (d and e): Scatterplot of (d) SSS-TA and (e) SSS-nTA. The black line is the approximate line ($R^2 = 0.94$, TA = (50.46 ± 0.06) × SSS + (528.04 ± 2.00) (Ave. ± SE)).

番号 : 1 作成者 : Frédéric Gazeau  日付 : 2025/09/22 18:49:37
Figure captions should not include undefined acronyms, please detail all (e.g. Sea Surface Salinity (SSS), Total alkalinity (TA) etc..

番号 : 2 作成者 : Frédéric Gazeau  日付 : 2025/10/08 16:04:02
Missing units on the left plots (or add them to the caption)

[revised manuscript text omitted]

番号 : 1 作成者 : Frédéric Gazeau   日付 : 2025/09/22 19:02:30

番号 : 2 作成者 : Frédéric Gazeau   日付 : 2025/09/22 19:02:55
What about organic alkalinity?????

作成者 : Tatsuki Tokoro   タイトル : ノート注釈   日付 : 2025/10/10 15:49:11
Thank you for your valuable comment. As you rightly pointed out, it was an oversight not to address organic Alkalinity in the analysis of land-derived TA. The sentences about organic Alkalinity has been added at the end of this paragraph.
The added sentences state that, based on referential values for organic Alkalinity from mainland China (no Japanese data could be found), it was considered negligible under the conditions in this study. It also adds that it could have a significant impact on TA budget analysis in environment with lower salinity or abundant wetland such as Hokkaido -which were not included in this study. (Line 302-312)

番号 : 3 作成者 : Frédéric Gazeau   日付 : 2025/10/09 11:19:26
Nonetheless

[revised manuscript text omitted]

---

## Author Response (AR3)

Dear Dr. Gazeau,

Thank you for your comments. We appreciate you taking the time to provide your valuable feedback. Below is our response to your comments. Please note that we have not provided responses to simple grammatical corrections. We hope this revised manuscript meets the standards of Journal of *Biogeosciences*.

Line 24
Not clear, why don't you mention lowered omega levels?

The sentence and the following one have been revised as shown below for clarity. "The supply of riverine water had a minor effect on pH but contributed to coastal acidification, as indicated by a decrease in the calcification index ($\Omega_{arg}$, the aragonite saturation state) by 0.09 ± 0.01 over the past 20 years, even after accounting for the buffering effect of riverine Total Alkalinity, which reduced the overall decrease by approximately 71%.". (Line 24-26)

Line 24
not clear wht means this index

The explanation has been added as mentioned in the above response.

Figure 2
Define

We have added the explanation as "Empirical Orthogonal Function (EOF)". (Line 221)